# scLENS: data-driven signal detection for unbiased scRNA-seq data analysis

Hyun Kim[1], Won Chang [2], Seok Joo Chae[1,3], Jong-Eun Park [4], Minseok Seo[5] &
Jae Kyoung Kim [1,3] ✉

High dimensionality and noise have limited the new biological insights that can be discovered in scRNA-seq data. While dimensionality reduction tools have been developed to extract biological signals from the data, they often require manual determination of signal dimension, introducing user bias. Furthermore, a common data preprocessing method, log normalization, can unintentionally distort signals in the data. Here, we develop scLENS, a dimensionality reduction tool that circumvents the long-standing issues of signal distortion and manual input. Specifically, we identify the primary cause of signal distortion during log normalization and effectively address it by uniformizing cell vector lengths with L2 normalization. Furthermore, we utilize random matrix theory-based noise filtering and a signal robustness test to enable data-driven determination of the threshold for signal dimensions. Our method outperforms 11 widely used dimensionality reduction tools and performs particularly well for challenging scRNA-seq datasets with high sparsity and variability. To facilitate the use of scLENS, we provide a user-friendly package that automates accurate signal detection of scRNA-seq data without manual time-consuming tuning.

Single-cell sequencing, which encompasses genomic, transcriptomic, proteomic, and epigenomic sequencing, is a prominent tool used across biological research areas[1–4]. In particular, single-cell RNA sequencing (scRNA-seq) data have been widely employed in diverse downstream analyses, including clustering analysis for identifying cell-type-specific phenotypes[5–7], trajectory analysis for exploring cell differentiation and development[8,9], ligand-receptor network analysis for investigating cell-to-cell communication[10,11], and gene-oriented analysis for gene regulatory network reconstruction[12,13]. Despite this widespread use and utility, analysis of scRNA-seq data remains challenging due to skewed and biased data distribution, stochastic dropout, and technical noise[14–19].

Skewness and bias in scRNA-seq data distribution can result in overemphasis of highly expressed genes or prevalent cell types, leading to potential inaccuracies in downstream analysis, such as missing rare cell types[18,19]. To mitigate skewness and bias in data, log normalization has been widely used for data preprocessing (Fig. 1 left)[15,16,20,21]. However, conventional log normalization can introduce false variability by amplifying gaps between zero and non-zero values and not uniformly normalizing genes across different expression levels[22,23]. These side effects potentially lead to false discoveries, such as the misclassification of cells[22]. While alternative methods have been developed, including Pearson residuals and count-based factor analysis models[22–24], a recent study surprisingly demonstrated that the performance of these approaches is sub-optimal compared to log normalization[15]. This highlights the emerging necessity for alternative approaches to log normalization.

Another difficulty in analyzing scRNA-seq data stems from its sparsity, with a substantial portion of data entries being zeros, which can potentially exceed 90% in some data[17–19]. These zeros can represent

[1]Biomedical Mathematics Group, Pioneer Research Center for Mathematical and Computational Sciences, Institute for Basic Science, Daejeon 34126, Republic of Korea. [2]Division of Statistics and Data Science, University of Cincinnati, Cincinnati, OH 45221, USA. [3]Department of Mathematical Sciences, KAIST, Daejeon 34141, Republic of Korea. [4]Graduate School of Medical Science and Engineering, KAIST, Daejeon 34141, Republic of Korea. [5]Department of Computer and Information Science, Korea University, Sejong 30019, Republic of Korea. ✉e-mail: jaekkim@kaist.ac.kr

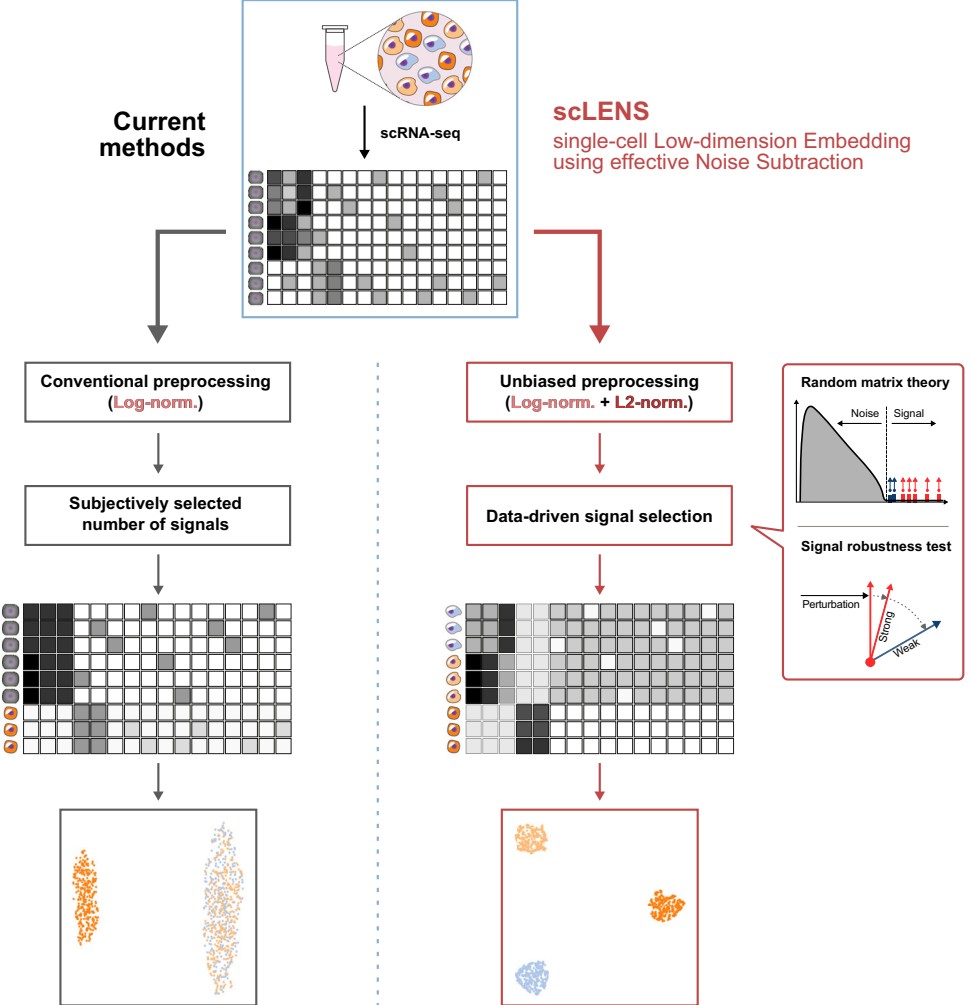

**Fig. 1 | Overview of scLENS (single-cell Low-dimensional embedding using the effective Noise Subtract).** Current dimensionality reduction methods employ log normalization for data preprocessing, which can distort signals in data due to the high level of sparsity and variance between cells (left). They then reduce the data using various dimensionality reduction algorithms. However, during this process, the majority of current methods require the user's decision to set a threshold to differentiate signals from noise. Due to signal distortion and manual signal

selection, current methods often fail to capture the high-dimensional data structure. In contrast, scLENS can prevent signal distortion by incorporating L2 normalization (right). Furthermore, scLENS uses random matrix theory-based noise filtering and signal robustness test-based filtering to automatically select signals without manual selection. As a result, scLENS can perform accurate dimensionality reduction without user bias.

either the actual absence of mRNA or a loss of information due to dropout events[17,25]. To address the biologically irrelevant zeros introduced by dropouts, various imputation methods have been widely implemented[25–29]. Specifically, they either replace zeros with non-zero values[27,29,30] or employ zero-preserving imputation techniques to retain most zeros as signals[25,26]. However, such imputation approaches can lead to misinterpretation of data since they have the risk of biasing data, leading to false signal[17,28,30,31]. Another approach is feature selection to reduce the number of zeros in the data by selecting highly variable genes, but this approach often leads to a significant loss of information contained in non-selected genes[18,19,25].

Additionally, scRNA-seq data is generally high dimensional and noisy[14,19,32]. To filter out noise and retain only the low-dimensional true biological signals, various dimensionality reduction methods (DR methods) have been developed. However, most DR methods require subjective user decisions to set the threshold that differentiates signal from noise (Fig. 1 left), introducing manual determination of dimension reduction[8,20,21,33]. Such user subjectivity can reduce the reliability and reproducibility of the results, leading to inconsistent outcomes across various analyses and thus compromising the objectivity of the

findings[34]. To address this limitation, recent studies have focused on leveraging the inherent noise structure to remove noise without relying on subjective user input[32,35,36]. For instance, Aparicio et al. introduced a denoising algorithm based on random matrix theory (RMT), which automatically distinguishes signal from noise in scRNA-seq data[32]. However, this method did not exhibit substantial performance improvements when compared to existing approaches[37]. This underscores that not only effective noise filtering, but also accurate preprocessing and removing low-quality signals are required to obtain high-quality biological signals from scRNA-seq data.

In this study, we have developed scLENS (single-cell Low-dimension Embedding using effective Noise Subtraction) which captures biological signals accurately and automatically (Fig. 1 right). ScLENS comprises modified log normalization for unbiased preprocessing and RMT-based noise filtering and post-filtering of signals for data-driven signal detection (Fig. 1 right). Specifically, we found that the most popular preprocessing method, log normalization, distorts signals due to its inability to uniformize cell vector lengths. To address this, we incorporated an additional L2 normalization step following log normalization. RMT-based noise filtering was then

applied to the normalized data to automatically identify biologically meaningful signals without user subjective choice. Among these signals, signals that were robust to binary sparse perturbation of data were selected, effectively removing low-quality signals caused by dropouts without data manipulation. By overcoming the long-standing challenges in scRNA-seq analysis and the limitations of conventional methods, scLENS outperforms existing downstream analysis tools based on various preprocessing and DR methods[8,20,21,23,27,32,33,38–40]. scLENS especially excels in analyzing complex data featuring high levels of sparsity, cell's total gene counts (TGC) variance, and non-binary information. By automating accurate biological signal detection, eliminating the need for labor-intensive parameter tuning, scLENS boosts downstream analysis of scRNA-seq data, fostering a deeper understanding of complex biological processes.

## Results

### The application of conventional log normalization results in excessive detection of the signals

Most popular analysis tools for scRNA-seq data, including Seurat, Scanpy, and Monocle3, employ log normalization with various scaling factors, typically greater than 1000 by default, as a preprocessing step to address bias and skewness in data[8,20,21]. They then reduce the dimensionality of data to capture true biological signals by filtering out noise[16,18]. During this process, setting a threshold for distinguishing signal from noise is crucial. However, in most cases, this decision is left to the user[8,20,21,33]. To circumvent this subjective choice, we employed a recently developed noise filtering method based on the random matrix theory (RMT-based noise filtering)[32]. RMT-based noise filtering provides a data-driven threshold that distinguishes biological signals from random noise in the data (Fig. 2a). Specifically, we first preprocessed the data using log normalization. Then, by multiplying the log-normalized data by its transpose, we calculated the cell similarity matrix (Fig. 2b). Subsequently, using the Eigenvalue Decomposition (EVD) algorithm, a comprehensive set of eigenvalues of the cell similarity matrix was obtained (Fig. 2c). These values were fitted to a Marchenko-Pastur (MP) distribution, a universal distribution of eigenvalues obtained from random matrices. Eigenvalues conforming to the MP distribution are considered to be noise from a random matrix, while those deviating from the MP distribution, surpassing the threshold of the Tracy-Widom (TW) distribution (vertical line in Fig. 2c), are considered to be potential biological signals[32,41]. Using 33 detected signal eigenvalues and their corresponding eigenvectors, i.e., signal vectors (matrix shown in Fig. 2c), low-dimensional data can be obtained. However, the 2D embedding obtained with UMAP on the low-dimensional data failed to capture the high-dimensional data structure precisely (Fig. 2d). We suspected this inaccuracy arises from the excessive number of detected signals, which is overly abundant for differentiating merely three clusters (Fig. 2c).

### L2 normalization prevents signal distortion due to conventional log normalization

To find out the cause of this excessive signal detection, we constructed a 2000 by 2000 pure noise random matrix with elements drawn from a Poisson distribution with a mean of 2 (Fig. 2e). When log normalization was applied to the matrix, all eigenvalues of its cell similarity matrix (Fig. 2f) followed the MP distribution, i.e., no signal was detected (Fig. 2g). This can be understood from the cell similarity matrix (Fig. 2f), whose elements are defined as the inner product between two cell vectors, with each vector being an array of normalized gene expression levels from a single cell. As a result, the diagonal elements in this matrix represent the square of the lengths of the cell vectors, which are comparable to each other due to log normalization (Fig. 2f). This general structure allows off-diagonal elements in the matrix to be interpreted as the directional similarity between cell vectors. Thus, off-diagonal elements close to zero in the matrix (Fig. 2f) indicate no

substantial directional similarity between cells, explaining the absence of signal (Fig. 2g). Next, we concatenated the dense random matrix (Fig. 2e) and the sparse binary matrix with the size 2000 by 8000 to reflect the high sparsity of scRNA-seq (Fig. 2h). When log normalization was applied to this sparse random matrix, no signal was detected again, indicating that signal distortion does not occur even when data shows high sparsity (Fig. 2i, j).

To further reflect the bias in the cell's TGC in scRNA-seq data, we divided the first 400 rows of the dense part of the random sparse matrix by two, resulting in a cell group of low TGC (Fig. 2k). Such bias in the TGC, i.e., the differences in sequencing depth, was expected to be removed by the log normalization. However, unexpectedly, ~400 signals, surpassing the TW threshold, were detected (Fig. 2m). As the number of detected signals matches the number of cells with low TGC, we hypothesized that cells with low TGC are associated with artificial signals. We noticed that 400 diagonal elements of the cell similarity matrix corresponding to cells with low TGC were much larger than the other diagonal elements (Fig. 2l, red box), unlike the cell similarity matrix of the dense random matrix (Fig. 2f) and the sparse random matrix (Fig. 2i). This means that the lengths of 400 cell vectors with low TGC are much longer than that of the other 1600 cell vectors. As a result, the inner products with the long 400 cell vectors (Fig. 2l, red box) became larger than those with the other cell vectors (Fig. 2l, outside of the red box). In short, the cell similarity matrix no longer accurately reflects the directional similarity among the cell vectors. In particular, the increased inner products with the low TGC vectors, caused by the exaggerated lengths of the low TGC vectors (Fig. 2l), created an artificial directionality toward the low TGC vectors. This explains why the 400 eigenvalues of the cell similarity matrix surpassed the TW threshold (Fig. 2m). Such distortion of the signal became higher as higher levels of sparsity and TGC variance were introduced to random matrix (Supplementary Fig. 1).

The signal distortion occurs because log normalization fails to introduce uniformity to cell vector lengths when a matrix is highly sparse and has a bias in TGC, which is typical of scRNA-seq data[14,16,18,19]. This is surprising because during the first step of log normalization, library size normalization eliminates cell-specific bias in data by dividing gene expression in each cell by its TGC (i.e., step normalizes the cell vector lengths)[15,16,18,21]. However, we found that this library size normalization is disrupted by the follow-up gene scaling step because it over-amplifies non-zero values in genes containing many zeros, especially in cells with low TGCs (Supplementary Fig. 2).

Thus, to uniformize the lengths of cell vectors, we added L2 normalization after gene scaling. Although L2 normalization is a very simple approach, it removed all artificial signals effectively (Fig. 2n, o, Supplementary Fig. 1). Next, we tested whether L2 normalization can improve the low-dimensional embedding in Fig. 2d. When L2 normalization was applied to the log-normalized data, cell vector lengths became identical (Fig. 2p), and thus the number of detected signals was considerably reduced, from 33 (Fig. 2c) to 6 (Fig. 2q). This yielded 2D embedding (Fig. 2r), which more accurately portrays the high-dimensional data structure compared to previous one (Fig. 2d). However, overlaps between clusters still existed in 2D embedding, limiting resolution (Fig. 2r).

### Signal robustness test filters out low-quality signals due to non-biological zeros

Despite considerable improvement after L2 normalization, we still observed some overlap between the sub-clusters (Fig. 3a–c). We hypothesized that this sub-optimal result stems from the noise associated with the biologically irrelevant zeros because some zeros in scRNA-seq data are caused by stochastic dropout events rather than biological zeros. To handle biologically irrelevant zeros, various imputation methods have been developed[25–29]. However, every imputation technique unavoidably alters the original data, potentially

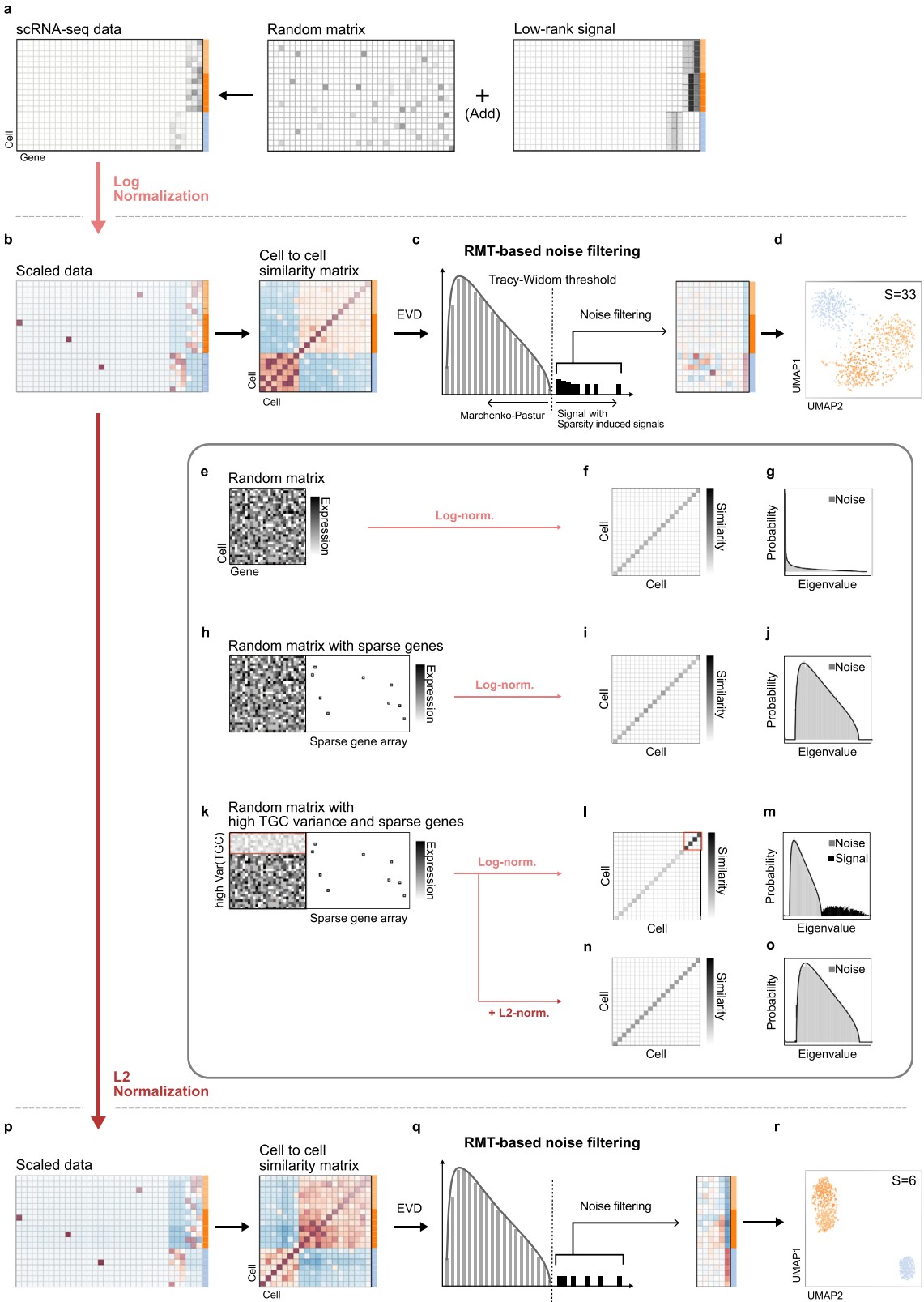

leading to the misinterpretation of the biological information contained within the data[17,28,30,31]. Thus, we developed an alternative method that preserves the original data while filtering out low-quality signals contaminated by biologically irrelevant zeros.

We found that the low-quality signals mainly stem from low expression of genes that do not share a strong common expression pattern across cells (Supplementary Fig. 3). Thus, they are expected to be susceptible to slight perturbation of data, which can mask the spurious correlations of sparse genes. To introduce slight perturbations to the original data, we generated a binary (0 or 1) random matrix with a sparsity level of 0.97 or greater and added it to the original count matrix (Fig. 3d). We then quantified how much the signal vectors (Fig. 3e) are perturbed by calculating the absolute inner product between all signal vectors from the unperturbed data and all

**Fig. 2 | The distortion of log normalization for data with high sparsity and variation in TGC can be corrected by L2 normalization. a** ScRNA-seq data can be viewed as a random matrix perturbed by a low-rank signal matrix. **b** After log normalizing the scRNA-seq data, a cell-to-cell similarity matrix was obtained by multiplying the normalized data matrix with its transpose. **c** Eigenvalues of the cell-to-cell similarity matrix are classified as noise-associated eigenvalues (gray bars), which lie in the Marchenko-Pastur distribution, and signal-associated eigenvalues (black bars), which surpass the Tracy-Widom threshold. By utilizing the signal eigenvalues and their corresponding signal vectors, the low-dimensional data was reconstructed. **d** When UMAP was applied to the reduced data to create a 2D embedding, it failed to accurately represent the high-dimensional data structure. **e** To investigate the source of the inaccuracy, we made a pure noise random matrix with elements drawn from a Poisson distribution with mean 2 (Poisson (2)). **f, g** When applying log normalization to scale the data, its cell similarity matrix had diagonals with similar magnitudes (**f**) and no signals (**g**). **h** A sparse random matrix was created by concatenating the dense random matrix of (**e**) with a sparse binary matrix. **i, j** After log normalization, no signal was detected from the cell similarity matrix. **k** To describe cell-to-cell heterogeneity in total gene count (TGC), values in the top rows (red box) of the dense part of the random matrix of (**h**) were halved. **l** In this case, even after log normalization, diagonal entries of the cell similarity matrix corresponding to the reduced top rows of the data (red box) were bigger than the others. **m** As a result, artificial signals were detected. **n, o** With additional L2 normalization, the cell similarity matrix had diagonals with the same magnitudes (**n**), and no signals were detected (**o**). **p** L2 normalization was additionally applied to the log-normalized data (**b**). **q, r** This reduced the number of signal-associated eigenvalues from 33 to 6 (**q**) and improved the 2D embedding (**r**).

eigenvectors from the perturbed data (Fig. 3f (i)). Next, we constructed the column-wise maximum vector of the inner product matrix (Fig. 3f (ii)), whose $i$-th component represents the inner product between the $i$-th signal vector and its most similar eigenvector from perturbed data. We obtained multiple column-wise maximum vectors by repeating the process with different perturbation matrices and then used their mean vector to indicate the stability of signal vectors (Fig. 3f (iii)). In particular, a large $i$-th component of the mean vector indicates that the $i$-th signal vector is robust against the data perturbation (Fig. 3f (iii), red arrows). Utilizing only these three robust signals (Fig. 3f (iii), red box) among six signals detected using RMT-based signal filtering (Fig. 3b) enabled accurate distinction of three clusters without any overlaps between them in the 2D embedding (Fig. 3g). This result underscores that selecting the correct number of signals is essential for the successful downstream analysis.

### scLENS (single-cell Low-dimensional Embedding using effective Noise Subtraction)

By integrating log normalization and L2 normalization into scRNA-seq data preprocessing, along with implementing signal detection using RMT-based noise filtering and a signal robustness test, we developed a dimensionality reduction tool, scLENS (single-cell Low-dimensional Embedding using effective Noise Subtraction). To facilitate the use of scLENS, we provide a user-friendly computational package that automates dimensionality reduction, thus bypassing the need for labor-intensive and time-consuming parameter tuning (see Supplementary Information for manuals).

We evaluated the performance of scLENS on real data using ZhengMix data[42], which consists of purified peripheral blood mononuclear cells (Fig. 3h (i)). This dataset has been utilized for various benchmarking studies since the data include true labels, but it is challenging to classify the cell types due to high sparsity and variation in the TGC[43,44]. Upon applying conventional log normalization to the data, T-cell subtypes were not clearly distinguished in the 2D embedding constructed from 84 detected signals detected by RMT-based noise filtering (Fig. 3h (ii) dashed circle). RMT-based noise filtering detected a reduced number of 42 signals after applying L2 normalization following log normalization, leading to the further refinement of the 2D embedding (Fig. 3h (iii) dashed circle). This embedding was more improved by selecting 13 robust signals from 42 signals detected by RMT-based noise filtering, using signal robustness test-based filtering (Fig. 3h (iv) dashed circles). While this result is automatically obtained via scLENS without any parameter tuning, the result is comparable with the best result of ZhengMix data obtained from massive parameter tuning with various DR methods (Supplementary Fig. 4)[44].

### scLENS excels at handling sparse data with high TGC variance
Next, we benchmarked scLENS with the other 11 popular packages with their default settings (see Supplementary Table 1 for details). Among these, well-known packages like Seurat, Scanpy, and Monocle3 employ log normalization for preprocessing and Principal Component

Analysis (PCA) with 50 principal components (PCs) by default for DR. Unlike these methods, ParallelPCA (Horn's parallel analysis) of PCA-tools automatically selects PCs based on their statistical significance against those of randomized data[45,46]. Similar to ParallelPCA, Randomly[32] also automatically selects the signals based on RMT and employs log normalization as preprocessing. On the other hand, DR methods implementing matrix factorization (ACTIONet)[33], random projection (SHARP)[38], and autoencoder (scDHA and scVI)[27,39] rather than PCA generally exclude the gene scaling step in log normalization during the preprocessing. Furthermore, scDHA[37] and SHARP[38] do not even use the library size normalization step, which is the first step of log normalization. We also examined ZINB-WaVE[40], employing a model-based DR method, and scTransform[23], using a model-based normalization method, as alternatives to log normalization.

To benchmark scLENS against other DR methods, we utilized simulation data generated by scDesign2[47], which produces simulated data with true labels. After training on immune cell data[6], we simulated approximately 60,000 simulation cells of ten T-cell subtypes. By subsampling ~3000 cells each from 60,000 simulated cells, we generated 13 datasets with different sparsity levels and the coefficient of variation (CV) of TGC.

From the 13 test datasets, we selected five datasets with similar CV values of TGC, but varying sparsity levels, to evaluate the impact of sparsity on the package's performance. For the data with the lowest sparsity, scLENS achieved the highest silhouette score (SIL score) (Fig. 4a, left). After scLENS, three packages—Scanpy, Seurat, and Monocle3—demonstrated overall good performances, all employing log normalization (Fig. 4a, left). Unlike these three packages that use a default of 50 PCs, ParallelPCA, which automatically selects PCs, showed slightly lower performance than them, but overall, it also showed a good performance. Next, scTransform, which utilizes model-based normalization, along with ACTIONet and scVI, both using log normalization without gene scaling, exhibited intermediate performance (Fig. 4a, left). Following these packages, Randomly, which employs log normalization as preprocessing and automatically selects PCs, showed below-average performance. Two packages, scDHA and SHARP, which use only log transformation during preprocessing and model-based DR (ZINB-WaVE), showed lower performance levels (Fig. 4a, left). As sparsity increases, the SIL score of all packages decreases (Fig. 4a, left). Nevertheless, the decrease in scLENS's performance was less than that of the other 11 packages (Fig. 4a, left).

The SIL score is a distance-based metric, so it can be sensitive to noise and outliers in the data. Thus, we used an alternative metric, the element-centric similarity (ECS), which measures the similarity between the clustering result obtained using hierarchical clustering and the ground truth label to evaluate a given DR method's performance (see "Method" section for details). In terms of ECS, scLENS achieved the highest performance at the lowest sparsity level, which is consistent with previous results evaluated by the SIL score (Fig. 4a, right). In addition, as sparsity levels increased, scLENS demonstrated minimal degradation in ECS, whereas the other DR methods

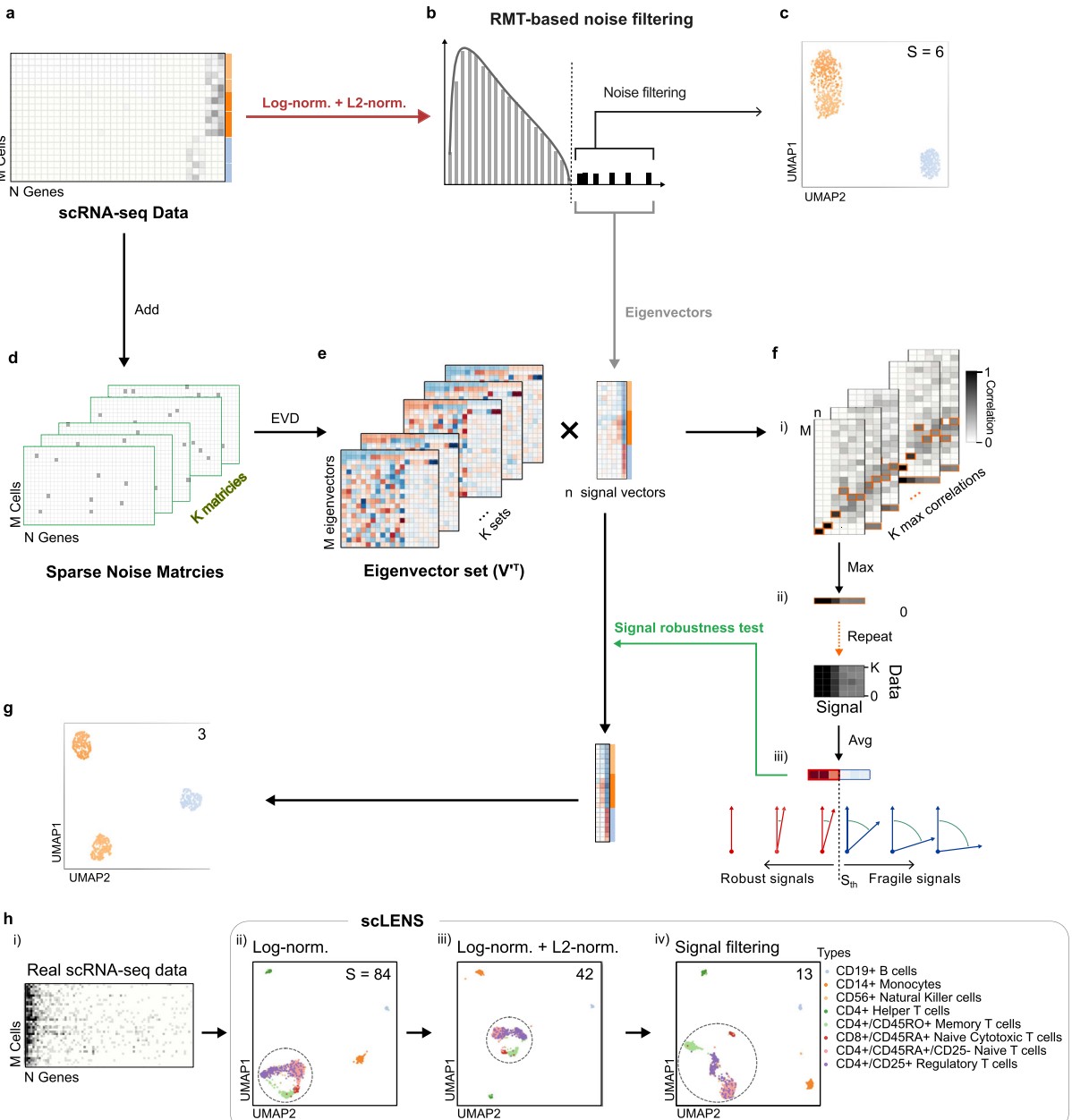

**Fig. 3 | Low-quality signals due to stochastic dropout can be filtered using a signal robustness test. a–c** Even when L2 normalization with log normalization was applied to the data (**a**), the six detected signals (**b**) led to the 2D embedding with limited resolution (**c**). **d** To filter out the low-quality signals sensitive to the slight perturbation of data, we generated K perturbed datasets by adding binary and sparse random matrices to the original count matrix. **e** We then calculated K eigenvector sets from the K perturbed datasets' similarity matrices. **f** To quantify the sensitivity of signals, we computed the correlation (i.e., absolute inner product) matrices (i) of the signal eigenvectors (**b**) and perturbed eigenvectors (**e**). Their column-wise maximum vectors were then obtained (ii) and averaged (iii). The high

value of the average vector means that signal vectors (red arrows) were robust to data perturbation (iii). **g** When three robust signals were used for 2D embedding, the accurate distinction of three sub-clusters was obtained. **h** When log normalization was applied to ZhengMix data (i), which are characterized by a high sparsity and CV of TGC, 2D embedding showed considerable overlap between T-cell subtypes (ii). With additional L2 normalization, 2D embedding was slightly improved (iii). After filtering out low-quality signals with signal robustness test, yielding 13 signals, 2D embedding demonstrated clear separations between T-cells subtypes (iv).

experienced significant declines in ECS (Fig. 4a, right). The more substantial performance degradation observed in DR methods using a fixed number of PCs compared to scLENS is attributable to their inability to account for the reduced biological information as sparsity increases. Furthermore, compared to these packages using a fixed number of PCs, Randomly and ParallelPCA, designed to select the PCs automatically, showed lower performance, suggesting their suboptimal effectiveness in identifying signals in data (Fig. 4a). On the other hand, scLENS's capability to identify the optimal number of

signals, which decreases from 22 to 18 with increased sparsity level, resulted in its 2D embedding providing a more distinct separation of true cell types compared to the 2D embeddings from the three packages using a fixed 50 PCs by default (Fig. 4b). As a result, scLENS achieved the highest performance, in terms of ECS and SIL (Fig. 4a, b).

Next, to investigate the influence of variation in CV of TGC on the performance of DR methods, we selected five datasets with similar sparsity levels, but varying CVs in TGC. When CV of TGC was low, scLENS and Monocle3 achieved the highest performance, while the

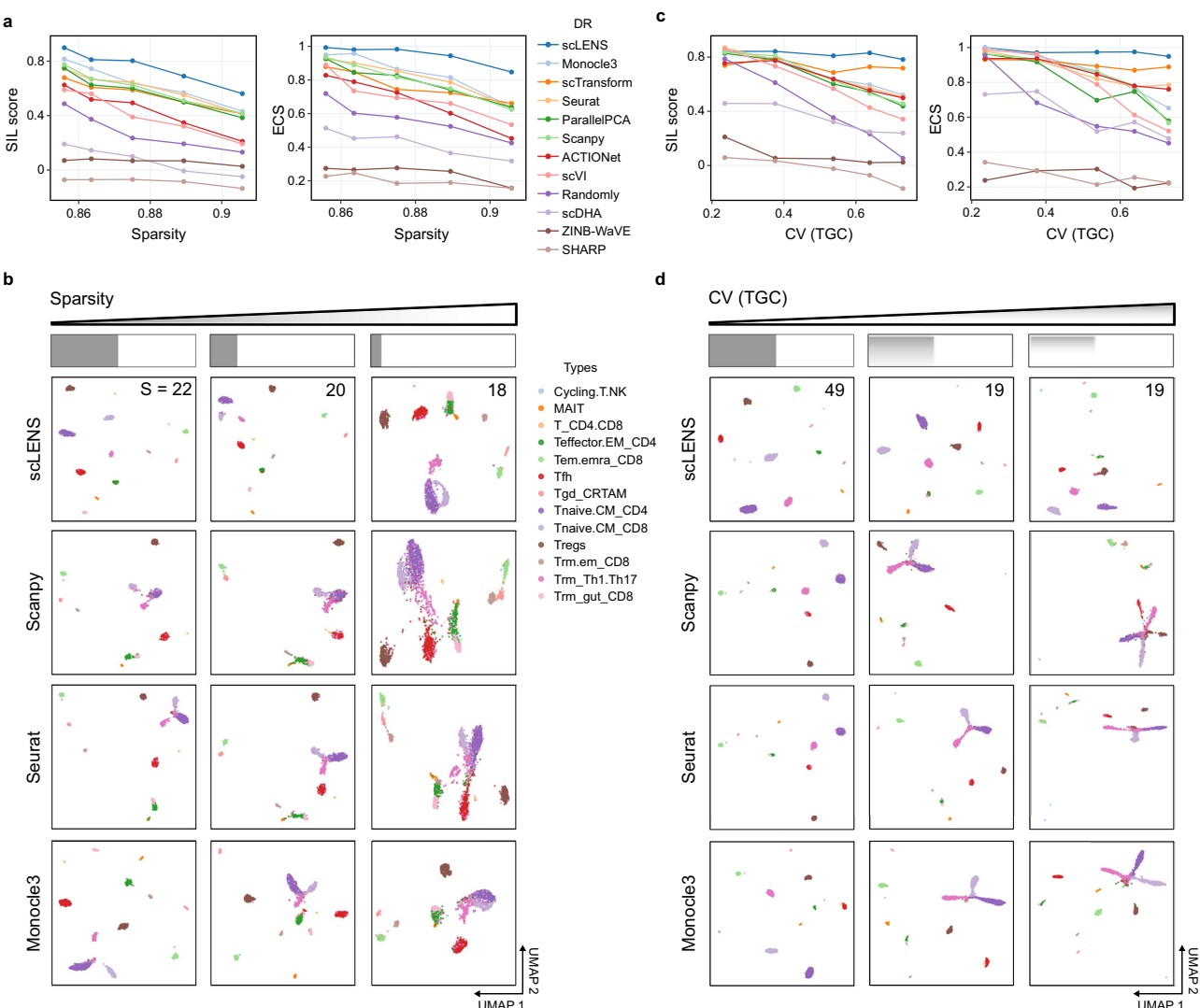

**Fig. 4 | Impact of sparsity level and TGC variation on the performance of dimensionality reduction (DR) methods. a** SIL scores (left) and ECS scores (right) for each DR method across datasets with different sparsity levels and TGC's CV values of around 0.3. **b** Influence of sparsity on inter-cluster distances in 2D embeddings generated by Scanpy, Seurat, Monocle3, and scLENS. As the sparsity level increased, scLENS detected reduced signals from 22 to 18, enabling a more distinct differentiation of the true cell types in 2D embedding compared to 2D embeddings of the others using fixed 50 PCs. **c** SIL scores (left) and ECS scores (right) for each DR method on datasets with varying CV of TGC and sparsity levels of around 0.84. **d** Effect of CV of TGC on cell point distribution in 2D embeddings produced by Scanpy, Seurat, Monocle3, and scLENS. Source data are provided as a Source Data file.

seven DR methods using log normalization, log normalization without gene scaling, and model-based scaling (scTransform) as preprocessing methods, demonstrated good overall performance (Fig. 4c). On the other hand, scDHA, ZINB-WaVE, and SHARP continued to display lower performance levels at the lowest CV of TGC (Fig. 4c). As CV of TGC increased, scLENS and scTransform showed no significant performance changes (Fig. 4c, right). In contrast, the performance of the seven DR methods, which use log normalization as preprocessing and a fixed number of signals by default, showed substantial decreases with an increasing CV of TGC (Fig. 4c right). Notably, when the CV of TGC was low, scLENS detected 49 signals, which is close to the default value of 50 PCs used by Scanpy, Seurat, and Monocle3. In this case, with 50 PCs by default, these three packages performed comparable to the scLENS by providing 2D embeddings showing the clear separation between true clusters. However, as the CV of TGC increased, the inter-cluster distances in the embeddings generated by Scanpy, Seurat, and Monocle3 became distorted, leading to significant overlaps between clusters (Fig. 4d). Conversely, scLENS maintained the clear separation between true clusters in its 2D embedding by effectively

reducing the number of detected signals from 49 to 19 as a CV of TGC increased (Fig. 4c, d). These results emphasize the importance of accurate signal selection, especially in data with high level of sparsity and CV of TGC.

## scLENS outperforms other DR methods on data with abundant non-binary information

We evaluated scLENS across a more diverse range of data types by combining 16 real and ten simulation datasets with the previous 13 simulated T-cell datasets (Supplementary Table 2). Specifically, to broaden our benchmarking on diverse types of data that encompass various cell types, we generated 10 additional simulation datasets from Tabula Muris data[48], obtained from various mouse tissues, with scDesign2[47]. Furthermore, to evaluate the performance of various benchmarking packages on real data, we included three real datasets, Koh[49], Kumar[50], and Trapnell data[51], whose cell labels were determined independently of the scRNA-seq assay to minimize evaluation bias towards particular analysis tools used in each study[43]. In addition, we used thirteen real datasets generated by mixing the Zheng data[42],

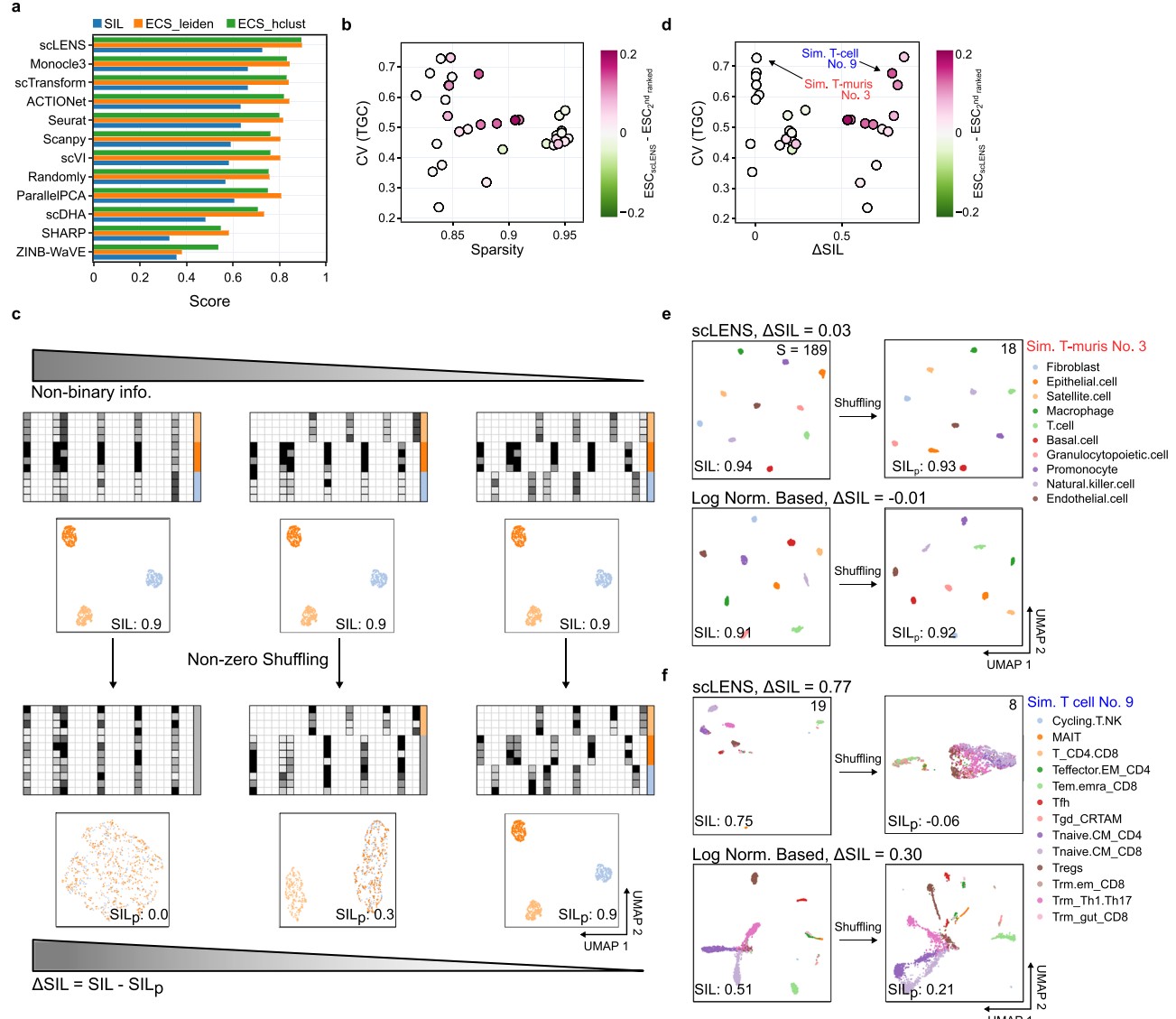

**Fig. 5 | Performance comparison of scLENS and other DR methods based on the amount of binary information. a** Benchmarking result of average SIL scores and ECS values shows that scLENS outperformed all other DR methods **b** Relative performance of scLENS (i.e., the difference between its ECS and the highest ECS recorded by the other 11 DR methods), according to sparsity and CV of TGC. As sparsity increases, the relative performance of scLENS increases, while no correlation is found between CV of TGC and relative performance. **c** As non-binary information derived from the magnitude variances in non-zero values in data decreases, the shuffling effect in non-zero values becomes weaker. This can be quantified with smaller ΔSIL values, which are differences in SIL scores of 2D

embeddings obtained by scLENS before and after shuffling non-zero values. **d** scLENS outperformed the other DR methods when both CV of TGC and ΔSIL were high. **e** When scLENS and Monocle3 have similar performance (the dataset with low ΔSIL and high CV in (**d**)), their embeddings are minimally affected by shuffling, indicating that the dataset contains a high proportion of binary information **f** When scLENS outperforms Monocle3 (the dataset with high ΔSIL and high CV in (**d**)), the embedding of scLENS, but not Monocle3 is significantly disrupted by shuffling, indicating that the dataset contains a high proportion of non-binary information. Source data are provided as a Source Data file.

which contains eight pre-sorted blood cell types, while adjusting the number of cells and the subpopulation ratios.

For the extended datasets, scLENS outperformed all other DR methods in terms of ECS scores based on both hierarchical clustering and graph-based clustering as well as SIL score (Fig. 5a). Next, we compared the performance of scLENS and the other DR methods depending on the sparsity level, and CV of TGC of the data, in terms of ECS score obtained by applying hierarchical clustering (Fig. 5b). For each dataset, we calculated the difference between the ECS of scLENS and the highest ECS recorded by the other 11 DR methods, referred to as the relative performance of scLENS. As the sparsity of data increases, the relative performance of scLENS showed overall increases (Fig. 5b), consistent with our previous results based on simulated data

(Fig. 4a). On the other hand, there is no correlation between CV of TGC and the relative performance (Fig. 5b), in contrast to our analysis based on simulated data (Fig. 4c).

Next, we investigated why there is no correlation between the CV of TGC and relative performance. Generally, the data contains two types of clustering information: binary and non-binary information. The binary information comes from the indices (positions) of the zero-valued elements within the data matrix, while non-binary information stems from the different magnitudes in non-zero values. The non-binary information is expected to be more distorted by conventional log normalization compared to the binary information because the conventional log normalization can overly amplify the gap between zero and non-zero values and reduce the variance in the non-zero

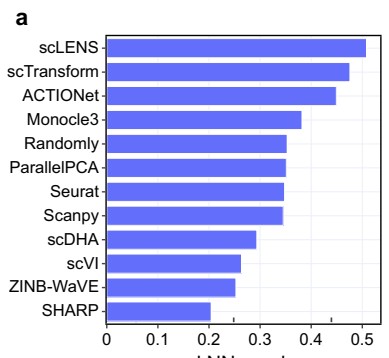
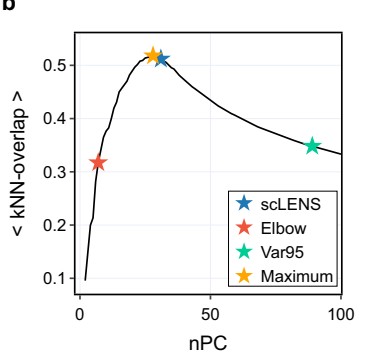
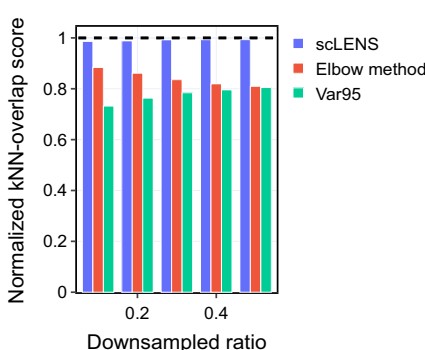

**Fig. 6 | Performance comparison of scLENS and other DR methods using the average kNN-overlap scores. a** Overall downsampling benchmarking results show that scLENS outperformed all other DR methods in terms of the average kNN-overlap score. **b** Three average kNN-overlap scores at three different numbers of PCs were obtained using scLENS (blue star), the elbow method (red star), and the 95% variance criterion (green star). scLENS identifies a near-optimal number of PCs, achieving an average kNN-overlap score close to the highest average kNN-overlap score (orange star), while the elbow method and the 95% variance criterion select a

small and large number of PCs, respectively, compared to the optimal number of PCs. **c** With increasing downsampling ratios, there is a slight improvement in the performance of the 95% variance criterion (green), while the performance of the elbow method (red) shows a decline. On the other hand, scLENS (blue) consistently selects the number of PCs almost close to the optimal, thereby achieving an average kNN-overlap score that closely aligns with the peak performance benchmark (black dashed line). Source data are provided as a Source Data file.

values[22] (Supplementary Fig. 6). Thus, we hypothesized that as the proportion of non-binary information decreases, the distortion of the log transformation becomes weaker and thus the relative performance of scLENS decreases. To test this hypothesis, we quantified the proportions of non-binary and binary information in the dataset. Specifically, we randomly shuffled the non-zero values to disrupt the non-binary information and then calculated the degree of change in SIL scores of 2D embeddings obtained by scLENS ($\Delta SIL = SIL - SIL_p$) (Fig. 5c, Supplementary Fig. 5). As the proportion of non-binary information decreases, the effect of shuffling in non-zero values decreases, and thus $\Delta SIL$ decreases (Fig. 5c).

Indeed, the relative performance of scLENS depends on $\Delta SIL$, measuring non-binary information (Fig. 5d). Specifically, when data contains mostly binary information and little non-binary information (i.e., low $\Delta SIL$), scLENS's relative performance is low regardless of CV of TGC (Fig. 5d). For example, in the simulated Tabula Muris data (Sim. T-muris No. 3 in Fig. 5d), the difference between 2D embeddings of scLENS before and after shuffling was barely noticeable (Fig. 5e). This indicates that the clustering for this data is mainly based on binary information. In such cases, scLENS and Monocle3 have no performance differences. On the other hand, for data with high non-binary information (i.e., high $\Delta SIL$) (Fig. 5d), scLENS outperforms the other methods. For instance, in simulated T-cell data (Sim. T-cell No. 9 in Fig. 5d), the 2D embedding of scLENS is completely disrupted by the shuffling of non-zero values (Fig. 5f top), indicating that the patterns in non-zero values are critical for the clustering. In contrast, shuffling shows a little disruption of 2D embedding by Monocle3 (Fig. 5f bottom). This indicates that the embedding of Monocle3 is mainly based on binary rather than non-binary information. This occurs because conventional log normalization exaggerates binary information, thereby causing an artificial reduction in the relative portion of non-binary information in the sparse data with a high CV of TGC. This explains a recent puzzling study reporting that dimensionality reduction involving log normalization on count data generates similar low-dimensional embeddings to those obtained from binarized data[52]. Taken together, when there is enough binary information in a dataset for clustering (easy case), scLENS and the DR methods based on conventional log normalization have similar performance (Fig. 5d, e). On the other hand, when non-binary information is critical for clustering (difficult case), scLENS outperforms the other tested DR methods (Fig. 5d, f).

## scLENS outperforms other DR methods in capturing local structure in data

So far, the performance evaluation has focused on the clustering and UMAP embedding performances of 12 packages using a limited number of real and simulated datasets with ground truths. To extend our analysis and diversify data types, we performed a downsampling benchmark approach using kNN-overlap scores, inspired by the study of Ahlmann-Eltze et al.[15]. For this analysis, we newly collected 15 deeply sequenced UMI count datasets and four read count datasets, each characterized by an average TGC exceeding 25,000 per cell[53–62] (Supplementary Table 2). These datasets were then downsampled to an average TGC of 5000 per cell, aligning with the typical sequencing depth of 10x genomics data. Subsequently, downsampled and original data were reduced in their dimensionality after applying 12 DR methods. We then evaluated the similarity between two kNN-graphs constructed from dimensionally reduced original and downsampled data using the average KNN-overlap score, which estimated each package's performance in capturing the original local complex structure from downsampled data (see "Methods" section for details).

scLENS outperforms the other 11 DR methods (Fig. 6a), similar to the result of the clustering performance benchmark (Fig. 5a). Along with scLENS, scTransform and ACTIONet, which used model-based normalization and log normalization without gene scaling for preprocessing, respectively, demonstrated overall good performances (Fig. 6a). Next, Monocle3, employing log normalization for preprocessing, showed competitive performance (Fig. 6a). Compared to the Monocle3, using a fixed number of 50 PCs, two packages, ParallelPCA and Randomly, which automatically detect signaling PCs, exhibited lower performance, indicating their ineffectiveness in identifying the optimal number of signals in data (Fig. 6a). Following them, the widely used three packages, Seurat, Scanpy, and scVI, which utilize log normalization and feature selection to select highly variable genes during preprocessing, showed sub-optimal performance (Fig. 6a). Consistent with the findings in the clustering benchmark, three packages, scDHA, SHARP, and ZINB-WaVE, demonstrated lower performances in terms of the average kNN-overlap as well (Fig. 6a).

Additionally, using the average kNN-overlap score, we compared scLENS's effectiveness in determining the optimal number of signals against two well-known methods: the elbow method and the 95% variance criterion. For this comparative analysis, we downsampled 19

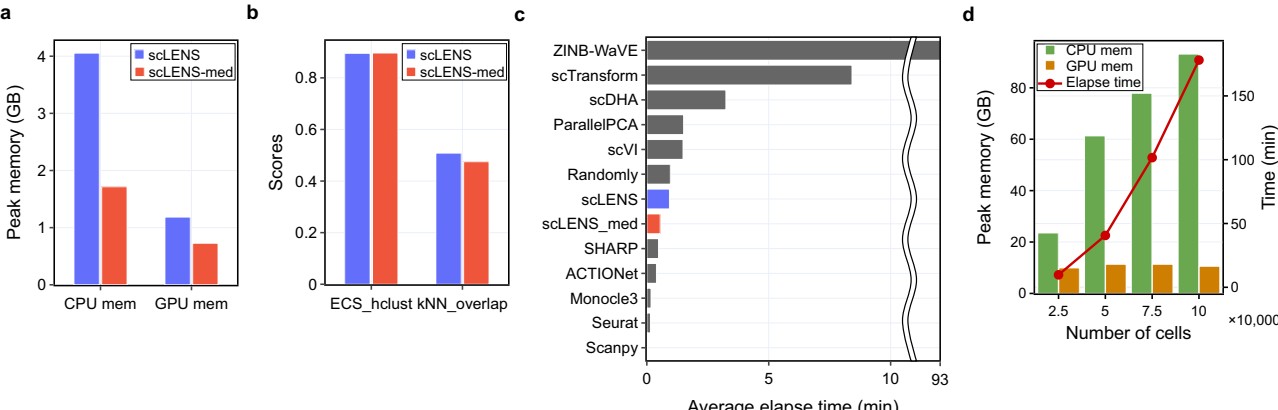

**Fig. 7 | The effectiveness of median scaling in terms of memory saving and speed performance. a** scLENS-med requires much less CPU and GPU memory compared to scLENS. **b** Comparison of scLENS and scLENS-med performance in terms of ECS and the average kNN-overlap score. **c** Although scLENS already demonstrates competitive speed performance with the other DR packages, scLENS-med approximately doubles its speed. **d** When scLENS-med is applied to large datasets with around 20,000 genes, the CPU memory (green bar) requirement increases as the number of cells increases from 25,000 to 100,000, while GPU memory (brown bar) requirements do not change significantly. scLENS-med takes ~3 h to analyze 100,000 cells and 20,000 genes (red line). Source data are provided as a Source Data file.

original datasets from a previous downsampling benchmark with ratios ranging from 0.1 to 0.5 and normalized them using log normalization with L2 normalization. We then reduced the normalized datasets to the top 100 PCs and generated 98 kNN graphs by varying the number of PCs from 3 to 100. By comparing these graphs to the reference kNN graph, constructed by applying scLENS to the original dataset, we computed the average kNN-overlap scores for each number of PCs. From these scores, three kNN-overlap scores, corresponding to three PC sets selected by scLENS (blue star in Fig. 6b), the elbow method (red star in Fig. 6b), and the 95% variance criterion (green star in Fig. 6b), were collected. Subsequently, we normalized these scores by dividing them by the maximum average kNN-overlap score (orange star in Fig. 6b) for each dataset to facilitate comparative analysis across different downsampling ratios (Fig. 6c). In this result, we found that, as the downsampling rate increases, the 95% variance criterion's performance increases, while the elbow method's performance decreases significantly (Fig. 6c). This implies that the 95% variance explained criterion often identifies more PCs as signals than the optimal number, while the elbow method typically selects fewer PCs as signals in comparison to the optimal number. In contrast, scLENS consistently chooses a number of signals close to the optimal number of signals, achieving performance that closely approaches the highest peak performance (Fig. 6c).

## Median scaling enhances the speed and memory efficiency of scLENS

Although scLENS demonstrates superior performance compared to other packages, it requires substantial memory due to its use of whole cells and genes that remain after quality control (QC). Post-QC, ~10,000 genes typically remain in most datasets, with some cases exceeding 18,000 genes (Supplementary Data 1). Furthermore, from these large-sized datasets, scLENS computes the complete sets of eigenvalues and eigenvectors to fit the MP distribution, which significantly increases scLENS's memory requirements. To reduce these memory requirements, we allocated only non-zero values and their indices in the data matrix to memory using SparseArrays.jl module. However, this approach loses its memory efficiency as a substantial increase in non-zero values occurs due to mean value subtraction during gene scaling. To address this, we modified the gene scaling step by replacing the subtraction value from the mean of each gene with their median (see Methods for details). Using the median scaling, we maintained a high level of sparsity even after the log normalization.

Indeed, scLENS with the median scaling (scLENS-med) requires lower memory than the original scLENS (Fig. 7a). Specifically, across the 39 datasets used in the clustering benchmark study, the CPU memory requirement was reduced by ~2.4 times, and GPU memory usage decreased by ~1.6 times on average (Fig. 7a). Despite the lower memory requirement, scLENS-med showed similar and slightly lower performance in terms of ECS and the average kNN-overlap score, respectively, compared to the original scLENS (Fig. 7b) across all 58 datasets used in previous benchmarking (Figs. 5 and 6). Moreover, scLENS-med was ~1.7 times faster than the original scLENS across the same datasets used in the memory performance analysis (Fig. 7c). This is noteworthy given that the original scLENS already had reasonable speed compared to the others due to utilizing GPU (Fig. 7c).

For four large datasets, each containing approximately 20,000 genes but varying in cell count from 25,000 to 100,000, scLENS-med requires reasonable memory and reasonable speed (Fig. 7d, e). That is, a dataset with 25,000 cells requires ~23GB of CPU memory and 20 min while processing 100,000 cells necessitates ~93 GB of RAM and 180 min (Fig. 7d green and Fig. 7e). This heavy CPU memory consumption occurs during preprocessing and generation of shuffled and perturbed data for the robustness test, and increases as the number of data matrix elements increases (Fig. 7d green). In contrast, the GPU memory consumption does not correlate with the number of cells in the data (Fig. 7d brown). This is because scLENS computes the same number of eigenvalues and eigenvectors as genes, usually lower than the number of cells in large datasets.

## Discussion

Although scRNA-seq has provided significant insights into complex biological systems, the inherent properties of these data, such as skewness, sparseness, and noise, have limited the information derivable from these datasets. Such issues stem from technical limitations, including amplification efficiency and stochastic dropout events[14–19]. Despite previous efforts to tackle the skewed and sparse nature of scRNA-seq data, current preprocessing steps can distort signals, and existing imputation methods are known to generate false signals[17,28,30,31]. Furthermore, DR methods for differentiating signals from noise rely on user input to select a threshold, introducing a subjective element to analysis. To address these long-standing challenges, we have developed scLENS. scLENS reduces signal distortion during the preprocessing step by incorporating L2 normalization and

provides a data-driven threshold to differentiate signals from noise by RMT-based noise filtering. In addition, scLENS removes low-quality signals arising from stochastic dropout events using a signal robustness test which does not require imputation. As a result, scLENS outperforms the popular elbow method and 95% variance criterion in detecting optimal number of signals. Importantly, scLENS exhibited superior performance for capturing biologically meaningful information from high-dimensional data compared to 11 popular packages (Figs. 4, 5, and 6). scLENS was especially effective when working on challenging data characterized by high sparsity, high variance between cells, and a substantial proportion of non-binary information (Figs. 4 and 5).

While log normalization is the most widely used preprocessing method, various alternative methods have been developed to confront its limitations[15,22–24]. However, a recent study by Ahlmann-Eltze et al. showed that the alternative methods, such as residual-based and latent gene expression transformation, underperformed compared to logarithmic transformation-based methods that include log normalization[15]. However, the conventional log normalization with a large scaling factor not only significantly amplifies the gap between zero and non-zero values[22] (Supplementary Fig. 6a), but also reduces the variance in large values (Supplementary Fig. 6b). As a result, current analysis tools employing log normalization exaggerate the binary information in data, making them less effective in capturing the non-binary information (Fig. 5f and Supplementary Fig. 6c). This explains why dimensionality reduction involving conventional log normalization generates similar low-dimensional embeddings even after binarizing data[52]. To circumvent data binarization, we set a scaling factor of 1 during the first step of log normalization. However, with a scaling factor of 1, log normalization can significantly distort the signals during gene scaling by reintroducing variability into the lengths of cell vectors (Fig. 2 and Supplementary Fig. 2). To counteract this, although Booeshaghi et al. recommended L1 normalization, it was ineffective in mitigating the compounding effect of sequencing depth, resulting in sub-optimal performances[15,63] (Supplementary Fig. 7). On the other hand, scLENS achieved the highest performance by employing an additional L2 normalization step as a remedy to correct signal distortion following log normalization (Fig. 2).

Since scRNA-seq data is noisy, it is essential to separate the biological signal from technical noise in scRNA-seq data[14,16,18,19,32,35–37]. One considerable strategy to accomplish this is applying the feature selection to select highly variable genes. However, the selection of genes based on high variance potentially compromise the local structure contained in low variance genes[18,19,25] (Fig. 6a). The other strategy is applying various DR methods, which extract low-dimensional signals from high-dimensional data. During this step, the dimension of these signals (i.e., the number of signals) should vary according to the datasets, as each dataset has its unique signal and noise structure. Nevertheless, the determination of optimal dimensions is usually left to the user's discretion, which can introduce potential subjectivity into downstream analysis results, thereby reducing the result's reliability. To remove user subjectivity in analysis, several optimal dimension decision methods have been developed. These include the elbow method and variance criterion, based on analyzing explained variance[64–69], and ParallelPCA, which focuses on statistical significance against randomized data[45,46]. However, these methods exhibited relatively lower performance than other packages, which might be due to the presumptions of normality, highly sensitive to outliers, and unable to eliminate user subjectivity completely[64,67] (Figs. 5 and 6). In particular, scLENS based on RMT, which removes user subjectivity, outperforms the elbow method and variance criterion (Fig. 6c) and ParallelPCA (Fig. 5a and Fig. 6a). Moreover, RMT-based noise filtering allows us to estimate the quantity of the information in the data, enabling us to assess the significance of signals[41].

Despite the advantages of the RMT-based approach, it necessitates the computation of all eigenvalues of the large cell-to-cell similarity matrix, requiring significant memory resources. Although we considerably reduced the memory demand of scLENS by replacing conventional gene scaling (z-score scaling) with median scaling, its operation on large data with whole genes still requires a computer with high specifications (Fig. 7). To circumvent this, one can use approximation of the spectral density of the similarity matrix, which estimates the range of the eigenvalues without calculating all eigenvalues when handling data obtained from tens of millions of cells[70]. Additionally, projecting all cells or genes using a model, such as a random forest learning method, trained on sub-sampled data could provide another alternative for analysis[7]. In the future, these strategies could be utilized to effectively manage the memory requirements associated with large scRNA-seq datasets, mitigating computational challenges while preserving the integrity of the analysis results.

Biologically irrelevant zeros in scRNA-seq data can hinder the accurate capture of biological signals in data. The most popular solution to resolve this challenge is replacing zeros with imputed values obtained from various imputation methods[25–29]. However, every imputation method inherently modifies the original data, and thus original signals can be compromised after imputation[17,28,30,31]. Indeed, a recent study found that original data performed better than using imputed data in the overall performance of downstream clustering and dimensionality reduction methods[17]. Due to the addition of a signal robustness test, scLENS does not require any imputation step or the modification of the original data to address biologically irrelevant zeros (Fig. 3). Additionally, since this signal robustness test does not require any assumption regarding signals, it can be integrated with the various DR methods, including linear DR methods such as PCA and nonlinear DR methods such as neural networks.

In summary, by addressing signal distortion induced by sparsity and effectively filtering out different types of noise, scLENS performed better than most popular scRNA-seq analysis packages, including those using signal selection options, such as ParallelPCA and elbow method (Fig. 5a and 6). Notably, scLENS showed superior performance for datasets even with high sparsity and high variance between the samples (Figs. 4 and 5b, d). Furthermore, as high sparsity and noise level are common characteristics of single-cell sequencing data[2–4], including the single-cell assay for transposase-accessible chromatin using sequencing (scATAC-seq) and single-cell proteomics data, scLENS potentially has broad applicability across single-cell sequencing analysis.

## Methods
### Preprocessing
Cells with fewer than 200 expressed genes and genes expressed in less than 15 cells were filtered out for quality control (QC). In addition, cells were discarded if the proportion of mitochondrial genes was larger than 5% to remove multiplets or low-quality cells[8,20,21,32].

After quality control, we apply the standard log normalization[8,20,21,32] while setting the scaling factor $L$ to be one.

$$X_{ij}^{\text{log-trans}} = \log\left(1 + \frac{X_{ij}^{\text{raw}}}{\sum_j X_{ij}^{\text{raw}}} L\right)$$
$$X_{ij}^{\text{gene-scaled}} = \frac{X_{ij}^{\text{log-trans}} - \mu_j}{\sigma_j} \tag{1}$$

for all $i = 1, \ldots, M$ and $j = 1, \ldots, N$, where $X_{ij}^{\text{raw}}$ is the original gene expression for the $i$-th cell and $j$-th gene, $M$ is the number of cells, $N$ is the number of genes, and $\sigma_j$ is the standard deviation of $j$-th gene's log-transformed expression levels. $\mu_j$ is the mean or median of $j$-th gene's log-transformed expression levels for scLENS and scLENS-med, respectively. We then applied L2 normalization to obtain an $M$ by $N$

data matrix whose $(i,j)$-th element is

$$X_{ij} = \frac{X_{ij}^{\text{gene-scaled}}}{\sqrt{\sum_j \left(X_{ij}^{\text{gene-scaled}}\right)^2}} \qquad (2)$$

## RMT-based noise filtering

After computing the cell similarity matrix ($G = XX^T/N$) of the normalized data $X$, we calculated eigenvalues and eigenvector matrix $V$ of $G$. Following the fitting procedure suggested by Aparicio et al.[32], MP distribution was fitted to the eigenvalue distribution of the cell similarity matrix $G$. By employing the parameters of the fitted MP distribution, we determined the TW threshold. This threshold represents the critical point, beyond which an eigenvalue has a 0.05 probability of being observed under the TW distribution of a random matrix. Subsequently, signal eigenvalues that exceed this threshold and their corresponding eigenvectors (i.e., signal vectors) $V^{\text{sig}}$ were identified and selected. Note that the product of the square root of $n$ signal eigenvalues and the corresponding eigenvectors $V^{\text{sig}}$ is mathematically equivalent to an $n$-dimensional PC score matrix calculated by multiplying $n$ PCs to the normalized data in PCA. Therefore, selecting $n$ PCs after PCA corresponds to detecting $n$ signals from the similarity matrix in scLENS.

## Signal robustness test

To test the robustness of signals obtained with RMT-based noise filtering, we perturbed the original data with a binary random matrix with a high sparsity level. The sparsity level for the perturbation matrix was determined through the following iterative procedure: For each iteration, we first generated a binarized matrix by replacing the nonzero values in the original data with ones. We then perturbed this matrix by adding a binary random matrix generated with the sparsity level given for the current iteration. After preprocessing on the binarized and perturbed matrix, we computed two sets of eigenvectors from their respective similarity matrices. We then calculated the minimum of their correlation. Starting from an initial sparsity level of 0.999, we gradually decreased the sparsity level for the perturbation matrix until this minimum correlation was less than the average correlation between two eigenvector sets derived from two different random matrices with the original data's sparsity level. Usually, the selected sparsity level was larger than 0.97.

Next, we generated 10 perturbed datasets by adding the perturbation matrix with the selected sparsity level to the original count matrix. After preprocessing on 10 perturbed datasets, 10 perturbed eigenvector sets of their similarity matrices ($V' = \{V'^{(1)}, V'^{(2)}, \ldots, V'^{(10)}\}$) were obtained. With 10 perturbed eigenvector sets, we subsequently computed 10 maximum absolute correlations ($C_{*i}$) of each signal vector ($\mathbf{v}_i = V_{*i}^{\text{sig}}$), which was obtained by applying the RMT-based noise filtering on the original data:

$$C_{ij} = \max_{1 \le k \le N} \left( \left| \left(V'^{(i)}\right)^T \cdot V^{\text{sig}} \right|_{kj} \right), \text{ where } i = 1, 2, \ldots, 10. \qquad (3)$$

We defined each signal's average of the 10 maximum correlations as its robustness value.

$$\text{Robustness of signal } j = \frac{1}{10} \sum_i C_{ij} \qquad (4)$$

Signals with a robustness value greater than 0.5 were selected as robust signals. Finally, we constructed reduced data with the signal eigenvalues and signal vectors corresponding to the selected robust signals.

## UMAP

We used UMAP[71] after reducing the dimension of data to get the 2D embeddings for the evaluation of performances. We employed the Julia package (UMAP.jl) using parameters with the number of nearest neighbors of 15 and a minimum distance of 0.01 to emphasize the local data structure. When applying UMAP, the metric used to calculate distances between data points was set to cosine similarity. This is a more appropriate metric for estimating the similarity of cell vectors' direction.

## Clustering

Two types of cluster assignments were used for the performance evaluation. The first type was obtained by implementing hierarchical clustering[72] on 2D embedding. The second type was obtained by applying the widely used graph-based clustering algorithm, the Leiden algorithm[73], to a shared nearest neighbors (SNN) graph. This graph was constructed from low-dimensional embedding generated by each package, using three parameters for SNN graphs: 20 neighbors, an edge cutoff threshold of 1/15, and cosine similarity as the distance metric. After performing multiple hierarchical and graph-based clustering with varied cut tree heights and resolution parameters, respectively, we selected the two types of cluster assignment that exhibited the maximum ECS similarities to the ground truth for each package. Using these selected two types of cluster assignments as the best clustering outcomes for each package, we evaluated the clustering performance of each package.

## Evaluation metrics

**Silhouette score.** The silhouette scores of each cell were calculated using the difference between the mean intra-cluster distance ($a$) and the mean closest-cluster distance ($b$)[74]. The silhouette score for the $i$-th data point representing a single cell is

$$s_i = \frac{b_i - a_i}{\max\{a_i, b_i\}} \qquad (5)$$

We utilized the average silhouette score (SIL score: $\langle s \rangle$) to evaluate the performance.

SIL score is in the range $[-1, 1]$. The best SIL score is 1. On the other hand, a SIL score of 0 indicates overlapping clusters and a negative SIL score usually means that cells were misclassified.

**ECS.** The existing clustering comparison metrics, such as an adjusted Rand index (ARI), Fowlkes-Mallows index (FM index), and normalized mutual information (NMI), have critical biases which undermine their usefulness[75]. For example, if one of the cluster assignments being compared has many clusters, the NMI value tends to be high[75]. Furthermore, the FM index tends to be high when one of the cluster assignments being compared includes a large cluster. ARI also has an unintuitive tendency when cluster assignments have a skewed cluster size[75]. These biases of existing metrics lead to a counterintuitive conclusion. In contrast, the element-centric similarity (ECS) can measure the similarity between two cluster assignments without such biases and provides an intuitive quantification of clustering similarity[75].

To calculate the ECS, two cluster-induced element networks were constructed from two cluster assignments ($l_\alpha, l_\beta$) being compared. These networks consist of edges connecting nodes that belong to the same cluster. Given these networks, two personalized PageRank (PPR) affinities ($f_{ij}^\alpha, f_{ij}^\beta$), which indicate the attribute of cell $i$ for another cell $j$, were obtained. Using these affinities, the ECS is defined as,

$$S_{\alpha\beta} = \frac{1}{N} \sum_{i=1}^{N} \left( 1 - \frac{1}{2d} \sum_{j=1}^{N} \left| f_{ij}^\alpha - f_{ij}^\beta \right| \right) \qquad (6)$$

We use the default value of the damping factor $d = 0.9$[75]. The best ECS is 1, and the worst ESC is 0.

**Average kNN-overlap score.** To estimate the overlaps between two kNN graphs constructed, respectively, from the original and down-sampled data, we initially counted the number of common edges that connect identical pairs of nodes. This count of common edges was then normalized by dividing $(n \times k)/2$, representing the maximum total number of edges in a kNN network, where $n$ is the number of cells and $k$ is the number of neighbors. Subsequently, by varying $k$ from 5 to 50, we computed 46 normalized counted numbers of edges, and defined their average as an average kNN-overlap score,

$$\frac{1}{46}\sum_{k=5}^{50}\left(\frac{n \times k}{2}\right)^{-1}\sum_{i,j=1}^{n}\frac{|\text{idx}_i \cap \text{idx}_j|}{2} \qquad (7)$$

where $\text{idx}_i$ is the list of nearest neighbors' indices of cell $i$. An average kNN-overlap score of 1 signifies that two compared graphs are identical, indicating complete overlap. Conversely, an average kNN-overlap score of 0 denotes the absence of any common edges between the two graphs, indicating no overlap.

**Benchmarking packages**

We compared 12 packages, including scLENS (Supplementary Table 1). We followed the standard processes with default values as suggested on the package websites and relevant papers. For the scVI package, we selected and used 10,000 genes using a function called 'highly_variable_genes' from Scanpy.

**Statistics & reproducibility**

To ensure the reproducibility of our signal detection method, we conducted a signal robustness test by performing ten perturbations and estimating the average angle change of each signal vector across these perturbations (Fig. 3). We measured the amounts of non-binary information using the ΔSIL scores, which were calculated by randomly shuffling the non-zero values of datasets with known ground truth labels (Supplementary Fig. 5).

For the benchmark study on the effects of data sparsity and high TGC (Total Gene Counts) variance on performance, we generated ~60,000 simulated immune cells. To minimize unintended effects due to sample size, we sampled ~3000 cells using weighted random sampling based on the probability weights of cells' TGC to create 13 simulated immune cell datasets from the pool of ~60,000 cells. In contrast, for the real ZhengMix datasets[42], to ensure a diverse range of cluster sizes and ratios between clusters, we subsampled and mixed cells of different types using weighted random sampling based on the probability weights of the target size of cell types.

To generate downsampled datasets, we utilized weighted random sampling with probability weights proportional to each cell's gene expression levels.

No data were excluded from the analyses. The Investigators were not blinded to allocation during experiments and outcome assessment.

**Versions of packages**

Version 1.0.0 of the scDesign2 package was used for generating simulation data. Versions of R and Python packages for benchmark studies are provided in Supplementary Table 1. R-based packages were run using R version 4.3.2, Python-based packages were run using Python version 3.11.5, and scLENS was built using Julia 1.8.5 and tested with Julia 1.10.0.

**Reporting summary**

Further information on research design is available in the Nature Portfolio Reporting Summary linked to this article.

## Data availability

The real datasets used in this study are publicly available and can be accessed through the following sources: Koh[49], Kumar[50], and Trapnell[51] datasets were obtained from the GitHub repository [https://github.com/markrobinsonuzh/scRNAseq_clustering_comparison]. Zheng datasets[42] were obtained from 10x Genomics datasets [https://www.10xgenomics.com/resources/datasets]. The Tabula Muris dataset[48] was obtained from the Tabula Muris Project [https://tabula-muris.ds.czbiohub.org/]. The immune cell dataset[6] was obtained from the Cross-tissue Immune Cell Atlas [https://www.tissueimmunecellatlas.org]. The mouse fibroblasts dataset[61] was obtained from ArrayExpress under the accession code E-MTAB-10148. The perinatal mouse hematopoietic stem cells dataset[62] was obtained from ArrayExpress under the accession code E-MTAB-13293. The fibroblast and HEK cells dataset[58] were obtained from ArrayExpress under the accession code E-MTAB-8735. The siRNA KnockDown dataset[61] was obtained from the GitHub repository [https://github.com/sandberg-lab/lncRNAs_bursting/tree/main/data]. The JM8 cells dataset[57] was obtained from Gene Expression Omnibus (GEO) database under the accession code GSE103568. The HEK293FT, K562, and human PBMC cells datasets[60] were obtained from ArrayExpress under the accession code E-MTAB-11467. The human brain cells dataset[54] was obtained from the Hemberg Lab repository [https://hemberg-lab.github.io/scRNA.seq.datasets/human/brain/#darmanis]. The scRNA-seq dataset with cells from mouse zygotes to blastocysts[53] was obtained from the Hemberg Lab repository [https://hemberg-lab.github.io/scRNA.seq.datasets/mouse/edev/#deng]. The mouse embryos dataset[55] was obtained from the Hemberg Lab repository [https://hemberg-lab.github.io/scRNA.seq.datasets/mouse/edev/#goolam]. The colorectal tumor cells dataset[56] was obtained from the Hemberg Lab repository [https://hemberg-lab.github.io/scRNA.seq.datasets/human/tissues/#li]. Source data for Figs. 4–7 and Supplementary Figs. 6–7 have been provided with this paper. A selection of the real datasets and all simulated datasets used in this study are available in the GitHub repository [https://github.com/Mathbiomed/scLENS] and archived at Zenodo[76]. Source data are provided with this paper.

## Code availability

The Julia codes for the scLENS, and the codes for the R and Python packages used in the benchmark study are available in the Mathbio GitHub: https://github.com/Mathbiomed/scLENS. The package version used for the analyses in the paper has been assigned a citable DOI through Zenodo (https://doi.org/10.5281/zenodo.10839592)[76].

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

## Acknowledgements

We thank SungHwan Bae of BSTAR ARTWORK for his assistance in preparing the figures. We also thank Life Science Editors for editing support. This work was supported by Institute for Basic Science IBS-R029-C3 (to J.K.K.), the National Research Foundation of Korea (NRF) grant funded by the Korean government (MSIT) (No. 2021R1A5A8032895 to M.S.S.), and the Charles Phelps Taft Research Center at the University of Cincinnati (no. M80941 to W.C.).

## Author contributions

H.K. and J.K.K. conceived of the study. H.K. and J.K.K. wrote the paper. H.K. developed the scLENS Julia package. H.K. performed data analysis with assistance from J.E.P and M.S.S. H.K., W.C., S.J.C., and J.K.K. discussed the scLENS method.

## Competing interests

The authors declare no competing interests.
