## [Peer Review File · Nature Communications]

scLENS: Data-driven signal detection for unbiased scRNA-seq data analysisEditorial Note: This manuscript has been previously reviewed at another journal that is not operating a transparent peer review scheme. This document only contains reviewer comments and rebuttal letters for versions considered at *Nature Communications*.

Reviewer #1 (Remarks to the Author):

The authors have addressed all of my comments, and I am happy to recommend the text for publication. The caveat is the code which I have been unable to review and I cannot vouch for it.

The only minor issue the legend for Fig 5b,c which are monochromatic.

Reviewer #1 (Remarks on code availability):

The authors did not provide a link to the code.

Reviewer #3 (Remarks to the Author):

I would like to first apologize to authors and editor for this review report being very late. I was first travelling, then sick, then overly busy trying to catch up, which caused me to fall behind with all kinds of tasks last month. I still haven't found the time to look through the revision with as much scrutiny as I would have liked --especially, I wanted to try the software on my own data--, but a short report might be better than an even later long one. So this here is what I have so far.

In my first report, I was very sceptical about the point of the whole method. I claimed that it does not matter whether one uses 30 or 50 PCAs, as long as it is neither too few, not excessively many. This claim was born out of my practical experience from using standard analysis methods. Not only me, but also the other reviewers had reservations on whether the presented benchmarks were sufficient to demonstrate that scLens offers actual benefit on real data. In response, the authors have now greatly expanded their benchmarks and also improved the description of details, and I have to say that this definitely made the manuscript much more convincing.

After studying all the new papers I have now realized a flaw in the logic of my original theoretical objections: My experience with using Seurat-like workflows taught me that the exact number of PCs has little influence on the outcome as long as one stays within a reasonable range. But this is precisely due to the normalization issue that the authors point out as their core result: Each PC is a mixture of signal and noise, and as the proportion of noise raises rather gradually, using a few more or a few less PCs does not make much of a difference. However, as the authors demonstrate, their extra normalization step changes precisely this and causes a clean unmixing of signal and noise so that each PCs is now either nearly only signal or nearly only noise. The more I think about it the more I have to agree that this is a really insightful and very useful result.

I had also, like Reviewer 1, reservations on the scalability of the approach, given that a full eigenspectrum seems needed. The authors have now improved their method (by introducing the median scaling) and demonstrates that this now makes analysis of large state-of-the-art data sets feasible. This was certainly a crucial addition to the manuscript.

Overall, concerning the fundamental objections I raised in my first review, I am happy to day that the authors rebuttal and revision has convinces me otherwise, and that I now see great merit in the work.

As stated, I regret that I have not found the time for a thorough reading of all the changes to the manuscript; I mainly studied the rebuttal. Therefore, I refrain from commenting on whether there are minor points that still need fixing or whether the manuscript's revised structure now presents the story well. I trust that the other reviewers will have covered this.

I only add two wishes that the authors may or may not implement:

- The new Figure 6b now shows how signal-to-noise ratio depends on the number of PCs taken if one uses the authors' extra L2 normalization. To illustrate how crucial that is, one could add a second line to the plot that shows the kNN overlap when using the standard process without the extra L2 normalization.

- I made one try running the software on my own data that I had available in the form of the output files of 10X/CellRanger. I tried to use the supplied script 'convert_to_jld', which did not work due to mismatching file names. I managed to edit and fix the file, but then failed again downstream. I do not want to blame the authors for the issue as supplying such conversion scripts is, in my experience, an uphill battle, as possible input formats keep changing. What I would have liked is, instead of a script, a written explanation describing the exact content that the JLD file should have. (It seems it should have one object called 'data' that is a sparse matrix genes x cells (or transposed?) with rownames but without column names.)

Alternatively, a Julia notebook might be handy that demonstrates reading in raw data from 10X files, bringing these into the required shape, explicitly performing the log normalization followed by L2 normalization and then handing over this matrix to the GPU-based eigen-solver, and then the resulting spectrum to the Marchenu-Pastur fitting function, which returns the number of eigenvectors to retain. Then the notebook could save the truncated eigenvector matrix and let the user continue with that using Seurat, Scanpy or whatever. The advantage of such a notebook over a black-box script like the 'scLENS.jl' file provided is that it allows the user to see all the steps that are easy and straight-forward, while the "difficult" parts of the code (the GPU-based eigen-solver and the Marchenu-Pastur fit) are done by function whose API is now clearly explained. That will make it much easier for advanced practitioners to implement the authors idea into existing workflows, and help users with input data in different formats to see how to get their data into the authors' procedure.

Reviewer #4 (Remarks to the Author):

The author revised the manuscript based on comments from the original Reviewer #2, particularly providing a detailed explanation of using random matrix theory to select an appropriate number of principal components, and adding relevant comparative validation with other methods. However, the following issues in this revised manuscript have not been adequately addressed.

1. The author has added comparisons and explanations about the ability of scLENS to capture local information (Major 1 in the original version of the comments), however, this cannot prove the author's claims that scLENS is suitable for solving signal distortion and avoid manual input. In addition, the author's response in Major 1 of the original comments failed to adequately explain the innovation of using random matrix theory to determine the number of principal components.

2. For Major 2 in the original version of the comments, the author has included an additional validation of the scLENS method by incorporating 15 unique molecular identifier (UMI) count datasets and 4 read count datasets. However, the author's response to the reviewer's request to include more real single-cell datasets for validating the scLENS method was inadequate. Instead of incorporating validation with real single-cell datasets covering various species, tissues, and platforms, the author only included validation using simulated datasets. This limits the persuasiveness of the results. In addition, the author did not provide an explanation or experimental validation of how the scLENS method captures more complex structures from single-cell data.

3. For a tool designed for single-cell data analysis, providing relevant biological insights is particularly important. However, the authors did not incorporate the reviewer's suggestion to add a section focusing on the biological insights that the scLENS method can achieve (Minor 10 in the original version of the comments).

Review response for “scLENS: Data-driven signal detection for unbiased scRNA-seq data analysis”

We thank the editor and the reviewers for their valuable feedback on our manuscript, constructive comments, and suggestions. In response to the reviewers’ comments, we have edited our manuscript. Below, we give our detailed responses to the comments and describe the changes in the manuscript. Our responses and the revised portions of the manuscript are highlighted in are denoted in blue.

Reviewer Comments

Reviewer #1 (Remarks to the Author):

The authors have addressed all of my comments, and I am happy to recommend the text for publication. The caveat is the code which I have been unable to review and I cannot vouch for it.

We thank Reviewer #1 for providing constructive feedback. Your valuable comments have helped us in enhancing the clarity of our manuscript.

The only minor issue the legend for Fig 5b,c which are monochromatic.

We have changed the color of the legend in Fig. 5b and c to be less confusing as follows:

- Fig. 5

Reviewer #1 (Remarks on code availability):

The authors did not provide a link to the code.

scLENS code has been uploaded to GitHub as <https://github.com/Mathbiomed/scLENS>

Reviewer #3 (Remarks to the Author):

I would like to first apologize to authors and editor for this review report being very late. I was first travelling, then sick, then overly busy trying to catch up, which caused me to fall behind with all kinds of tasks last month. I still haven't found the time to look through the revision with as much scrutiny as I would have liked --especially, I wanted to try the software on my own data--, but a short report might be better than an even later long one. So this here is what I have so far.

In my first report, I was very sceptical about the point of the whole method. I claimed that it does not matter whether one uses 30 or 50 PCAs, as long as it is neither too few, not excessively many. This claim was born out of my practical experience from using standard analysis methods. Not only me, but also the other reviewers had reservations on whether the presented benchmarks were sufficient to demonstrate that scLens offers actual benefit on real data. In response, the authors have now greatly expanded their benchmarks and also improved the description of details, and I have to say that this definitely made the manuscript much more convincing.

After studying all the new papers, I have now realized a flaw in the logic of my original theoretical objections: My experience with using Seurat-like workflows taught me that the exact number of PCs has little influence on the outcome as long as one stays within a reasonable range. But this is precisely due to the normalization issue that the authors point out as their core result: Each PC is a mixture of signal and noise, and as the proportion of noise raises rather gradually, using a few more or a few less PCs does not make much of a difference. However, as the authors demonstrate, their extra normalization step changes precisely this and causes a clean unmixing of signal and noise so that each PCs is now either nearly only signal or nearly only noise. The more I think about it the more I have to agree that this is a really insightful and very useful result.

I had also, like Reviewer 1, reservations on the scalability of the approach, given that a full eigenspectrum seems needed. The authors have now improved their method (by introducing the median scaling) and demonstrates that this now makes analysis of large state-of-the-art data sets feasible. This was certainly a crucial addition to the manuscript.

Overall, concerning the fundamental objections I raised in my first review, I am happy to day that the authors rebuttal and revision has convinces me otherwise, and that I now see great merit in the work.

As stated, I regret that I have not found the time for a thorough reading of all the changes to the manuscript; I mainly studied the rebuttal. Therefore, I refrain from commenting on whether there are minor points that still need fixing or whether the manuscript's revised structure now presents the story well. I trust that the other reviewers will have covered this.

Thank you for your thoughtful feedback and for recognizing the enhancements in our manuscript. Your insights have been pivotal in refining our work, and we are grateful for your acknowledgment of its improved clarity and merit.

I only add two wishes that the authors may or may not implement:

- The new Figure 6b now shows how signal-to-noise ratio depends on the number of PCs taken if one uses the authors' extra L2 normalization. To illustrate how crucial that is, one could add a second line to the plot that shows the kNN overlap when using the standard process without the extra L2 normalization.

We have previously shown that L2 normalization improves the local structure capturing performance in terms of the kNN-overlap score at the optimal number of PCs (SFig. 7). Thus, this seems not needed as it could make Fig. 6b complex. As the reviewer 3 suggests this comment is optional, we would like to keep the original Fig. 6b.

- I made one try running the software on my own data that I had available in the form of the output files of 10X/Cell Ranger. I tried to use the supplied script 'convert_to_jld', which did not work due to mismatching file names. I managed to edit and fix the file, but then failed again downstream. I do not want to blame the authors for the issue as supplying such conversion scripts is, in my experience, an uphill battle, as possible input formats keep changing. What I would have liked is, instead of a script, a written explanation describing the exact content that the JLD file should have. (It seems it should have one object called 'data' that is a sparse matrix genes x cells (or transposed?) with rownames but without column names.)

As the reviewer pointed out, the current file ('convert_to_jld.jl') might cause errors when the names and formats of the data differ from the provided data. To address this, we have added a description of the jld2 file in the scLENS manual and revised the code, allowing users to customize their input as follows:

- scLENS manual (README.md)

Output Information: After running the script `convert_to_jld.jl`, the resulting `.jld2` file includes a DataFrame representing single-cell RNA sequencing (scRNA-seq) count data. The first column in this DataFrame is designated for the cell ID. Each row corresponds to an individual cell, and each column represents a specific gene. You can customize the cell IDs and gene IDs as needed by altering the `barcodes.tsv.gz` and `features.tsv.gz` files, respectively.

Alternatively, a Julia notebook might be handy that demonstrates reading in raw data from 10X files, bringing these into the required shape, explicitly performing the log normalization followed by L2 normalization and then handing over this matrix to the GPU-based eigen-solver, and then the resulting spectrum to the Marchenu-Pastur fitting function, which returns the number of eigenvectors to retain. Then the notebook could save the truncated eigenvector matrix and let the user continue with that using Seurat, Scanpy or whatever. The advantage of such a notebook over a black-box script like the 'scLENS.jl' file provided is that it allows the user to see all the steps that are easy and straight-forward, while the "difficult" parts of the code (the GPU-based eigen-solver and the Marchenu-Pastur fit) are done by function whose API is now clearly explained. That will make it much easier for advanced practitioners to implement the authors idea into existing workflows, and help users with input data in different formats to see how to get their data into the authors' procedure.

We developed scLENS, which yields the output by simply receiving input data to emphasize that it requires a minimal number of user-defined parameters. However, as the reviewer pointed out, a conventional line-by-line execution approach does help in understanding the working principles of scLENS. Therefore, we are currently developing a Python version of scLENS that will offer these conventional execution approaches. We will add this version in the GitHub in the near future as soon as the Python version is ready.

Reviewer #4 (Remarks to the Author):

The author revised the manuscript based on comments from the original Reviewer #2, particularly providing a detailed explanation of using random matrix theory to select an appropriate number of principal components and adding relevant comparative validation with other methods. However, the following issues in this revised manuscript have not been adequately addressed.

We thank the reviewer to carefully check our responses to the comments of Reviewer 2.

1. The author has added comparisons and explanations about the ability of scLENS to capture local information (Major 1 in the original version of the comments), however, this cannot prove the author's claims that scLENS is suitable for solving signal distortion and avoid manual input. In addition, the author's response in Major 1 of the original comments failed to adequately explain the innovation of using random matrix theory to determine the number of principal components.

Because Random Matrix Theory (RMT)-based noise filtering can automatically select the number of signals using the universality of RMT, it was initially proposed as a solution to the challenge of manually selecting signals (i.e., principal components (PC)) (Aparicio, L. et al., *Patterns*, 2020). However, despite its innovativeness, this method has not been widely used due to its lower performance compared to other methods (Patrino, L. et al., *Briefings in Bioinformatics*, 2021). The cause of this suboptimal performance was poorly understood, but in this study, we found that it is attributed to signal distortion caused by conventional preprocessing, log normalization, and the high sensitivity of the RMT-based noise filtering to detect even low-quality signals. To address these issues, we merged L2 normalization and additional signal filtering based on a signal robustness test to the RMT-based noise filtering, yielding scLENS. Specifically, we found that the incorporation of L2 normalization into preprocessing successfully resolves the signal distortion caused by the log normalization (Fig. 2). Furthermore, with this correction of signal distortion and additional signal filtering based on the signal robustness test, the scLENS finds the optimal number of signals automatically (Fig. 6b), solving the long-standing issue of manual input in the selection of PCs. Importantly, scLENS shows higher performance than other packages on the real and simulated datasets, unlike the previous RMT-based noise filtering method (Randomly) without L2 normalization and noise filtering (Fig. 5a and 6a). Thus, we strongly believe that scLENS represents an innovative advancement in the RMT-based approach.

2. For Major 2 in the original version of the comments, the author has included an additional validation of the scLENS method by incorporating 15 unique molecular identifier (UMI) count datasets and 4 read count datasets. However, the author's response to the reviewer's request to include more real single-cell datasets for validating the scLENS method was inadequate. Instead of incorporating validation with real single-cell datasets covering various species, tissues, and platforms, the author only included validation using simulated datasets. This limits the persuasiveness of the results. In addition, the author did not provide an explanation or experimental validation of how the scLENS method captures more complex structures from single-cell data.

The 19 datasets used in our additional benchmark study (Fig. 6, 7b, and Supplementary Fig. 7), consisting of 15 UMI datasets and four read count datasets, are not simulated datasets generated using a simulator but are real datasets collected from various sources. Detailed information about these datasets is provided in Supplementary Table 2. However, we acknowledge that this data table was not appropriately referenced in the main text, and we apologize for any confusion caused by this. We have now revised the relevant part of the main text to refer to this data table correctly.

- Page 4 (results section)

For this analysis, we newly collected 15 deeply sequenced UMI count datasets and four read count datasets, each characterized by an average TGC exceeding 25,000 per cell⁵³⁻⁶² (Supplementary Table 2). These datasets were then downsampled to an average TGC of 5,000 per cell, aligning with the typical sequencing depth of 10x genomics data.

- Supplementary Table 2. Real and simulation datasets for benchmarking

Data set	Data type	# of data	Protocol	Description	Ref.
Kho	Real read count	1	SMARTer	FACS purified H7 human embryonic stem cells	[48]
Kumar	Real read count	1	SMARTer	Mouse embryonic stem cells, cultured with different inhibition factors	[49]
Trepnell	Real read count	1	SMARTer	Human skeletal muscle myoblast cells, differentiation induced by low-serum medium	[50]
Zhengmix4eq	Real UMI	1	10x	Mixture of purified peripheral blood mononuclear cells	[51]
Zhengmix4ueq	Real UMI	1	10x	Mixture of purified peripheral blood mononuclear cells	[51]
Zhengmix8eq	Real UMI	1	10x	Mixture of purified peripheral blood mononuclear cells	[51]
ZhengMix	Real UMI	10	10x	Mixture of purified peripheral blood mononuclear cells	[51]
Sim. Tabula muris	Simulated UMI	10	scDesign2	Data generated by training 10 cell types of Tabula muris data	[44, 47]
Sim. T cell	Simulated UMI	13	scDesign2	Data generated by training by 13 cell types of T-cells from Cross-tissue Immune Cell Atlas	[6, 44]
E-MTAB-10148	Real UMI	2	SmartSeq3	Primary mouse fibroblasts derived from the tail of a male adult mouse, F1 offspring of a C57 x CAST cross.	[61]
E-MTAB-13293	Real UMI	1	Smartseq3xpress	Perinatal liver Hematopoietic Stem Cells (HSCs) in Mus musculus	[62]
E-MTAB-8735	Real UMI	2	SmartSeq3	Fibroblast and HEK cells from mixed sample (Mus musculus, Homo sapiens)	[58]
SmartSeq3 siRNA_knockdown	Real UMI	1	SmartSeq3	Fibroblast siRNA Knockdown Data	[61]
mcSCRBseq	Real UMI	1	mcSCRB-seq	JM827 mouse embryonic stem cells	[57]
E-MTAB-11467	Real UMI	8	Smartseq3xpress	HEK293FT cells, K562 cells, and human peripheral blood mononuclear cells	[60]
Darmanis	Real read count	1	SMARTer	Human brain cells	[54]
Deng	Real read count	1	Smart-Seq2	Cells from mouse zygotes to late blastocyst stages, and adult liver cells, using crosses between CAST/Ei and C57BL/6 mouse strains	[53]
Goolam	Real read count	1	Smart-Seq2	Cells from mouse embryos at various stages (2-, 4-, 8-, 16-, and 32-cell)	[55]
Li	Real read count	1	SMARTer	Cells from colorectal tumors and their microenvironments	[56]

Moreover, in the benchmark studies using these 19 newly collected real datasets (Fig. 6, 7b, and Supplementary Fig. 7), we utilized the average kNN-overlap score proposed in the reference suggested by Reviewer 2's Minor comment 1 (Ahlmann-Eltze and Huber, *Nature Methods*, 2023). This score evaluates the similarity between the kNN graphs constructed from embeddings generated by each package using downsampled data and those constructed from high-quality data. As the kNN graph is utilized in a range of analyses, from basic analyses such as clustering to advanced analyses like pseudo-time estimation and trajectory analysis, it can be considered the fundamental structure that encodes the essential information of data (Ahlmann-Eltze and Huber, *Nature Methods*, 2023). Therefore, a package that achieves a high kNN overlap score can be considered effective at accurately capturing the complex fundamental structure of high-quality data from downsampled data. In this context, the high performance of scLENS in terms of the kNN-overlap score indicates its high effectiveness in capturing the original complex data structure from real scRNA-seq data (Fig. 6).

3. For a tool designed for single-cell data analysis, providing relevant biological insights is particularly important. However, the authors did not incorporate the reviewer's suggestion to add a section focusing on the biological insights that the scLENS method can achieve (Minor 10 in the original version of the comments).

Like the other methodological development studies for scRNA seq data analysis (Shahin Mohammadi et al., *Nat Commun*, 2020; Duc Tran et al., *Nat Commun*, 2021), we have decided to focus on the

outperformance of our methods compared to the previous methods rather than solving a specific biological problem. Furthermore, the reviewer 2 also said that this comment is optional. Thus, when transferring this study to Nature Communications, we provided the review plan to the editor team of Nature Communications, to make an agreement as follows:

“... the focus of our manuscript is on the methodological development and validation using benchmark data. Exploring biological insights, while valuable, falls beyond the scope of this manuscript. We will consider this exploration in subsequent studies. “

Furthermore, as we have specified in the review plan, we are in the process of conducting follow-up studies in which we apply scLENS to the *Drosophila* brain and Peripheral Blood Mononuclear Cell (PBMC) scRNA-seq data. These studies will showcase the practical utility of scLENS in biological research and maintain our engagement with the research community that utilizes our tool.

Reviewer #1 (Remarks on code availability):

I tried to follow the instructions for installing scLENS from the github page. It worked well with only a minor issue:

```
echo 'export PATH="$PATH:/home/users/julia-1.10.1/bin"' >> ~/.bashrc
source ~/.bashrc
```

it should be \$HOME not /home/users/

All of the files were named sim_Tcell_3 which does not seem ideal.

After trying to analyse the object it complains about not having the CUDA driver. I do not have nvidia drivers in my system and it looks like it runs on CPUs.

```
julia scLENS.jl data/sim_Tcell_3.csv.gz --true_label data/sim_Tcell_3_l.csv --out_dir out_dir --
device cpu --out_type julia --plot
```

```
parsed_args = parse_commandline() = Dict{String, Any}("arg1" => "data/sim_Tcell_3.csv.gz",
"scaling" => "mean", "device" => "cpu", "plot" => true, "out_type" => "julia", "true_label" =>
"data/sim_Tcell_3_l.csv", "out_dir" => "out_dir")
```

Parsed args:

```
arg1 => "data/sim_Tcell_3.csv.gz"
```

```
scaling => "mean"
```

```
device => "cpu"
```

```
plot => true
```

```
out_type => "julia"
```

```
true_label => "data/sim_Tcell_3_l.csv"
```

```
out_dir => "out_dir"
```

```
sim_Tcell_3.csv
```

```
preprocessing...
```

```
Inp_spec
```

```
data size: (3000, 12820), sparsity: 0.8860786852848637
```

```
After filtering>> data size: (2971, 11601), sparsity: 0.8733101010921668
```

```
scLENS...
```

```
Extracting matrices
```

```
Extracting Signals...
```

```
(Using cpu) number of signal ev: 63
```

```
Calculating noise baseline...
```

```
spth_: 0.06813231128264942
```

```
Calculating sparsity level for the perturbation...
```

```
ERROR: LoadError: CUDA driver not found
```

```
Stacktrace:
```

```
[1] error(s::String)
```

```
@ Base ./error.jl:35
```

```
[2] functional
```

```
@ ~/.julia/packages/CUDA/htRwP/src/initialization.jl:24 [inlined]
```

```
[3] task_local_state!()
```

```
@ CUDA ~/.julia/packages/CUDA/htRwP/lib/cudadvr/state.jl:77
```

```
[4] #alloc#1014
```

```
@ ~/.julia/packages/CUDA/htRwP/src/pool.jl:0 [inlined]
```

```
[5] alloc
```

```
@ ~/.julia/packages/CUDA/htRwP/src/pool.jl:421 [inlined]
```

```
[6] CuArray{Float32, 2, CUDA.Mem.DeviceBuffer}{::UndefinedInitializer, dims::Tuple{Int64, Int64}}
```

```
@ CUDA ~/.julia/packages/CUDA/htRwP/src/array.jl:74
```

```
[7] CuArray
```

```
@ ~/.julia/packages/CUDA/htRwP/src/array.jl:418 [inlined]
```

```
[8] adapt_storage
```

```
@ ~/.julia/packages/CUDA/htRwP/src/array.jl:740 [inlined]
```

```
[9] adapt_structure
@ ~/.julia/packages/Adapt/rkG93/src/Adapt.jl:57 [inlined]
[10] adapt
@ ~/.julia/packages/Adapt/rkG93/src/Adapt.jl:40 [inlined]
[11] #cu#1062
@ ~/.julia/packages/CUDA/htRwP/src/array.jl:805 [inlined]
[12] cu
@ ~/.julia/packages/CUDA/htRwP/src/array.jl:792 [inlined]
[13] _wishart_matrix(X::Matrix{Float32}; device::String)
@ Main ~/git/scLENS/scLENS.jl:585
[14] _wishart_matrix
@ ~/git/scLENS/scLENS.jl:578 [inlined]
[15] get_eigvec(X::Matrix{Float32}; device::String)
@ Main ~/git/scLENS/scLENS.jl:387
[16] get_eigvec(X::Matrix{Float32})
@ Main ~/git/scLENS/scLENS.jl:366
[17] scLENS(inp_df::DataFrame; device_::String, th::Int64,
l_inp::PooledArrays.PooledVector{String31, UInt32, Vector{UInt32}}, p_step::Float64,
return_scaled::Bool, obs_pt::String)
@ Main ~/git/scLENS/scLENS.jl:268
[18] main()
@ Main ~/git/scLENS/scLENS.jl:885
[19] top-level scope
@ ~/git/scLENS/scLENS.jl:941
in expression starting at /home/main/git/scLENS/scLENS.jl:941
```

Overall, I get the impression that an additional round of testing by external users would be helpful

Review response for “scLENS: Data-driven signal detection for unbiased scRNA-seq data analysis”

We thank the editor and the first reviewer for their valuable feedback and suggestions on our manuscript and some minor issues with the scLENS software. In response to these comments, we have updated our code and revised the instructions for scLENS. Below, we present our detailed responses to each comment and outline the modifications made to the installation instructions for scLENS available on the GitHub page. Our responses and the corresponding revisions are highlighted in blue.

Reviewer Comments

Reviewer #1 (Remarks to the Author):

I tried to follow the instructions for installing scLENS from the github page. It worked well with only a minor issue:

```
echo 'export PATH="$PATH:/home/users/julia-1.10.1/bin"' >> ~/.bashrc
source ~/.bashrc
```

it should be \$HOME not /home/users/

All of the files were named sim_Tcell_3 which does not seem ideal.

After trying to analyse the object it complains about not having the CUDA driver. I do not have nvidia drivers in my system and it looks like it runs on CPUs.

```
julia scLENS.jl data/sim_Tcell_3.csv.gz --true_label data/sim_Tcell_3_l.csv --out_dir out_dir -
-device cpu --out_type julia --plot
```

```
parsed_args = parse_commandline() = Dict{String, Any}("arg1" => "data/sim_Tcell_3.csv.gz",
"scaling" => "mean", "device" => "cpu", "plot" => true, "out_type" => "julia", "true_label" =>
"data/sim_Tcell_3_l.csv", "out_dir" => "out_dir")
```

Parsed args:

```
arg1 => "data/sim_Tcell_3.csv.gz"
```

```
scaling => "mean"
```

```
device => "cpu"
```

```
plot => true
```

```
out_type => "julia"
```

```
true_label => "data/sim_Tcell_3_l.csv"
```

```
out_dir => "out_dir"
```

sim_Tcell_3.csv

preprocessing...

Inp_spec

data size: (3000, 12820), sparsity: 0.8860786852848637

After filtering>> data size: (2971, 11601), sparsity: 0.8733101010921668

scLENS...

Extracting matrices

Extracting Signals...

(Using cpu) number of signal ev: 63

Calculating noise baseline...

spth_: 0.06813231128264942

Calculating sparsity level for the perturbation...

ERROR: LoadError: CUDA driver not found

Stacktrace:

[1] error(s::String)

@ Base ./error.jl:35

[2] functional

@ ~/.julia/packages/CUDA/htRwP/src/initialization.jl:24 [inlined]

[3] task_local_state!()

@ CUDA ~/.julia/packages/CUDA/htRwP/lib/cudadriv/state.jl:77

[4] #alloc#1014

@ ~/.julia/packages/CUDA/htRwP/src/pool.jl:0 [inlined]

[5] alloc

@ ~/.julia/packages/CUDA/htRwP/src/pool.jl:421 [inlined]

[6] CuArray{Float32, 2, CUDA.Mem.DeviceBuffer}{::UndefInitializer, dims::Tuple{Int64, Int64}}

@ CUDA ~/.julia/packages/CUDA/htRwP/src/array.jl:74

[7] CuArray

@ ~/.julia/packages/CUDA/htRwP/src/array.jl:418 [inlined]

[8] adapt_storage

@ ~/.julia/packages/CUDA/htRwP/src/array.jl:740 [inlined]

[9] adapt_structure

@ ~/.julia/packages/Adapt/rkG93/src/Adapt.jl:57 [inlined]

```

[10] adapt
@ ~/.julia/packages/Adapt/rkG93/src/Adapt.jl:40 [inlined]
[11] #cu#1062
@ ~/.julia/packages/CUDA/htRwP/src/array.jl:805 [inlined]
[12] cu
@ ~/.julia/packages/CUDA/htRwP/src/array.jl:792 [inlined]
[13] _wishart_matrix(X::Matrix{Float32}; device::String)
@ Main ~/git/scLENS/scLENS.jl:585
[14] _wishart_matrix
@ ~/git/scLENS/scLENS.jl:578 [inlined]
[15] get_eigvec(X::Matrix{Float32}; device::String)
@ Main ~/git/scLENS/scLENS.jl:387
[16] get_eigvec(X::Matrix{Float32})
@ Main ~/git/scLENS/scLENS.jl:366
[17] scLENS(inp_df::DataFrame; device_::String, th::Int64,
l_inp::PooledArrays.PooledVector{String31, UInt32, Vector{UInt32}}, p_step::Float64,
return_scaled::Bool, obs_pt::String)
@ Main ~/git/scLENS/scLENS.jl:268
[18] main()
@ Main ~/git/scLENS/scLENS.jl:885
[19] top-level scope
@ ~/git/scLENS/scLENS.jl:941
in expression starting at /home/main/git/scLENS/scLENS.jl:941

```

Overall, I get the impression that an additional round of testing by external users would be helpful

Thank you for your valuable feedback on the scLENS software. We have addressed each of your concerns as detailed below:

The installation instructions on our GitHub page have been updated to correct the path export command. The updated command is as follows:

```
echo 'export PATH="$PATH:$HOME/julia-1.10.1/bin"' >> ~/.bashrc
```

To improve the universality of our example command in the scLENS documentation, we've

made changes to utilize generic file names. The updated command is:

```
julia scLENS.jl data/your_dataset.csv.gz --true_label data/your_labels.csv --out_dir out_dir --  
device cpu --out_type julia --plot
```

Regarding the CUDA-related error messages when using a CPU, we have revised the scLENS code to address this issue. With the new updates, the software, when set to run on a CPU, will no longer produce these error messages.

We acknowledge the reviewer's comment that certain aspects of scLENS, such as the user experience, code organization, and hardware compatibility, need improvement. We will continue to integrate user feedback to enhance the functionality and usability of scLENS.